# Generation of an *Escherichia coli* strain growing on methanol via the ribulose monophosphate cycle

Philipp Keller [1,4], Michael A. Reiter[1,4], Patrick Kiefer[1], Thomas Gassler [1], Lucas Hemmerle[1,3], Philipp Christen [1], Elad Noor[2] & Julia A. Vorholt [1] ✉

Methanol is a liquid with high energy storage capacity that holds promise as an alternative substrate to replace sugars in the biotechnology industry. It can be produced from $CO_2$ or methane and its use does not compete with food and animal feed production. However, there are currently only limited biotechnological options for the valorization of methanol, which hinders its widespread adoption. Here, we report the conversion of the industrial platform organism *Escherichia coli* into a synthetic methylotroph that assimilates methanol via the energy efficient ribulose monophosphate cycle. Methylotrophy is achieved after evolution of a methanol-dependent *E. coli* strain over 250 generations in continuous chemostat culture. We demonstrate growth on methanol and biomass formation exclusively from the one-carbon source by $^{13}C$ isotopic tracer analysis. In line with computational modeling, the methylotrophic *E. coli* strain optimizes methanol oxidation by upregulation of an improved methanol dehydrogenase, increasing ribulose monophosphate cycle activity, channeling carbon flux through the Entner-Doudoroff pathway and downregulating tricarboxylic acid cycle enzymes. En route towards sustainable bioproduction processes, our work lays the foundation for the efficient utilization of methanol as the dominant carbon and energy resource.

Biotechnology is a key sector of the twentyfirst century and is expected to expand massively in the coming decades[1]. However, the currently applied biotransformation processes conflict with the production of human and animal foodstuffs as the production of the used raw materials predominantly relies on the exploitation of agricultural land. Therefore, alternative non-food and non-feed sources are required to replace sugar as the main substrate[2,3]. To this end, reduced one-carbon compounds, such as methanol, represent ideal base chemicals due to their high availability and energy density[4–6]. Furthermore, the steady increase in efficiency in the synthesis of methanol from the greenhouse gases $CO_2$ and methane will pave the way for sustainable biotechnological processes[7–14].

Organisms capable of using methanol as a growth substrate, referred to as methylotrophs, are abundant in nature; however, their biotechnological application is limited due to the lack of advanced genetic tools[2]. An alternative to relying on natural methylotrophs is to enable an already established platform organism, such as *Escherichia coli*, to metabolize methanol. The generation of synthetic methylotrophs has attracted considerable attention in the past few years and has mainly focused on the introduction of the ribulose monophosphate (RuMP) cycle for carbon assimilation due to its superior efficiency compared to alternative carbon assimilation pathways[15–30].

In *E. coli*, only three genes encoding a methanol dehydrogenase (*mdh*), a 3-hexulose 6-phosphate synthase (*hps*), and a 6-phospho 3-

[1]Institute of Microbiology, Department of Biology, ETH Zurich, 8093 Zurich, Switzerland. [2]Department of Plant and Environmental Sciences, Weizmann Institute of Science, Rehovot 7610001, Israel. [3]Present address: Laboratory for Environmental Biotechnology, Ecole Polytechnique Fédérale de Lausanne, Lausanne, Switzerland. [4]These authors contributed equally: Philipp Keller, Michael A. Reiter. ✉e-mail: jvorholt@ethz.ch

hexuloisomerase (*phi*) are lacking for a complete RuMP cycle[20]. Although the introduction of three enzymes seems straightforward, the implementation of a heterologous metabolic cycle is challenging as it requires the complete rewiring of the central metabolism of *E. coli*. In particular, carbon flux through the synthetic autocatalytic cycle must be tightly coordinated with its effluxes to achieve stable methanol assimilation[31]. One way to optimize the RuMP cycle is to introduce targeted mutations[26]. However, identifying and adequately modifying the operation of the metabolic network as a whole is difficult due to insufficient knowledge about enzyme properties and regulation. An alternative approach is adaptive laboratory evolution (ALE), which can lead to substantial metabolic changes towards a particular trait[18,32–36]. To achieve synthetic methylotrophy, a high selection pressure towards more efficient methanol assimilation is required during the ALE process. Initial dependence on methanol conversion can be successfully achieved by rational design of strains that are forced to build at least a portion of biomass precursors from the one-carbon source[15,17,19,29,30,37]. Because such methanol-dependent strains are only able to grow on a certain multi-carbon substrate in the presence of methanol, they represent ideal starting points for ALE towards a fully methylotrophic strain.

This strategy was pursued by different groups by engineering methanol-dependent strains that require, for example, glucose[16], gluconate[29], xylose[19] or ribose[19] as co-substrates for growth. However, these strains were constrained by an incomplete RuMP cycle that prevented evolution towards methylotrophy. This problem was recently approached by engineering a methanol-dependent strain with an incomplete RuMP cycle, evolving it, fixing the RuMP cycle at a later stage and further evolution until growth with methanol was observed[18]. In another work, Kim et al. transformed *E. coli* into a synthetic methylotroph by engineering the reductive glycine pathway[38]. Growth on methanol, however, was slow with a doubling time of about 54 h.

In a previous study, we identified methanol-dependent strains with a complete RuMP cycle[30]. A particularly promising strain contained two deletions: One in the triose phosphate isomerase (*tpiA*) that interrupted gluconeogenesis and abolished growth on pyruvate in the absence of methanol. The second mutation blocked the formaldehyde detoxification pathway by the removal of the *S*-(hydroxymethyl)glutathione dehydrogenase (*frmA*) to ensure high levels of formaldehyde, the one-carbon entry point of the RuMP cycle[20].

In this study, we show that a synthetic methylotroph can be evolved in a single long-term evolution experiment. The introduction of only two mutations (Δ*frmA*, Δ*tpiA*) together with the heterologous expression of three genes is sufficient to generate an *E. coli* strain with a complete RuMP cycle that grows on methanol after about 250 generations in a continuous chemostat culture. The evolved strain builds its entire biomass from the reduced one-carbon compound, as we demonstrate by metabolic tracer experiments and grows at a doubling time of about 8 h. Furthermore, we use a multi-omics approach and biochemical assays to characterize the synthetic methylotrophic *E. coli* strain as well as its evolutionary trajectory. The synthetic methylotroph and its analysis provides a valuable starting point for microbial conversion of methanol into value-added compounds and for applications in industrial biotechnology.

## Results

### Rewiring of central metabolic fluxes is required for growth on methanol

In this study, we aimed to generate an *E. coli* strain that is able to grow on methanol as the sole source of carbon and energy via the RuMP cycle. As a starting point, we selected the methanol-dependent strain Δ*frmA*Δ*tpiA*, which expresses the genes *mdh*, *hps*, and *phi* from a heterologous plasmid system. The strain requires methanol for growth and proved promising for evolution towards a methylotrophic lifestyle due to a complete set of RuMP cycle enzymes and methanol

incorporation into RuMP cycle metabolites[30]. Consistent with flux balance analysis (FBA), half of fructose 6-phosphate (F6P) was formed from methanol and dihydroxyacetone phosphate (DHAP) originated exclusively from methanol, while metabolites downstream of glyceraldehyde 3-phosphate (GAP) in glycolysis were generated from pyruvate, the second carbon source that was used as a growth substrate[30]. To evaluate the metabolic adaptations required for achieving growth solely on methanol, we modeled the central metabolism of *E. coli* by flux variability analysis (FVA), which in contrast to FBA, provides information regarding the overall solution space that could result in growth on methanol during evolution. Specifically, we compared the predicted distributions of metabolic flux during growth on methanol and pyruvate as co-substrates (Fig. 1a) and on methanol alone (Fig. 1b). To allow for a direct comparison of the fluxes, the target growth rate was set to $0.2\,h^{-1}$ in both conditions, a value similar to the growth rate of the native methylotroph *Bacillus methanolicus* at 37 °C[39]. Furthermore, for growth on pyruvate and methanol, the analysis was designed to reflect the situation in vivo, where pyruvate, a common substrate of *E. coli*, is more readily available to the cell than methanol due to its more efficient metabolization. To this end, the upper bound for the methanol uptake was set to the minimal amount of methanol uptake required ($0.6\,mmol\,gCDW^{-1}\,h^{-1}$) to achieve the target growth rate in the presence of excess amounts of pyruvate. In the case of growth on methanol alone, the methanol uptake rate was left unbounded. For growth on pyruvate together with methanol, the highest metabolic fluxes occurred through the tricarboxylic acid (TCA) cycle, while only a small fraction of the total fluxes originated from the RuMP cycle. In contrast, the predicted flux distribution during growth on methanol as the sole carbon source was markedly different. The fluxes through the TCA cycle were reduced to a minimum by a factor of 18, while the fluxes in the RuMP cycle increased on average 14-fold. Furthermore, the Entner–Doudoroff pathway, which was not active under mixed growth conditions, was predicted to be essential for the synthesis of more oxidized metabolites such as pyruvate. Overall, the altered metabolic fluxes confirmed that a complete restructuring of the central metabolism of *E. coli* is required to enable growth on methanol. Due to the number of the metabolic adaptations needed, a targeted engineering approach appeared too complex. For this reason, we chose an adaptive laboratory evolution experiment as the most feasible strategy to establish a fully methylotrophic lifestyle for the strain.

### Long-term evolution of the methanol-dependent strain Δ*frmA*Δ*tpiA*

To convert the methanol-dependent *E. coli* strain Δ*frmA*Δ*tpiA* into a fully methylotrophic organism, we performed long-term laboratory evolution. The strain was evolved under continuous conditions in a chemostat in which the dilution rate of the culture defined its growth rate (Fig. 2a). To ensure a high selection pressure towards increased methanol incorporation, methanol was present in excess (500 mM) and pyruvate at lower concentration (20 mM) in the chemostat feed which resulted in limiting concentrations of pyruvate (< 0.01 mM) in the growth vessel (Fig. 2c). Additionally, the feed medium was supplemented with 0.1 mM isopropyl-β-D-thiogalactopyranoside (IPTG) for heterologous expression of *mdh*, *hps* and *phi*. During the first ~150 days, which corresponded to about 90 generations based on the selected dilution rate, the optical density of the culture remained rather stable. Then, to further increase the selection pressure towards a higher methanol uptake, we gradually lowered the pyruvate concentration in the feed medium first to 10 mM and ultimately to 5 mM. While the pyruvate concentration in the growth vessel was already limiting with 20 mM of pyruvate in the feed medium, the reduction to 5 mM should lower the amplitude of the pyruvate fluctuations that arise when a droplet of feed medium enters the growth vessel. As expected, the optical density of the culture proportionally decreased

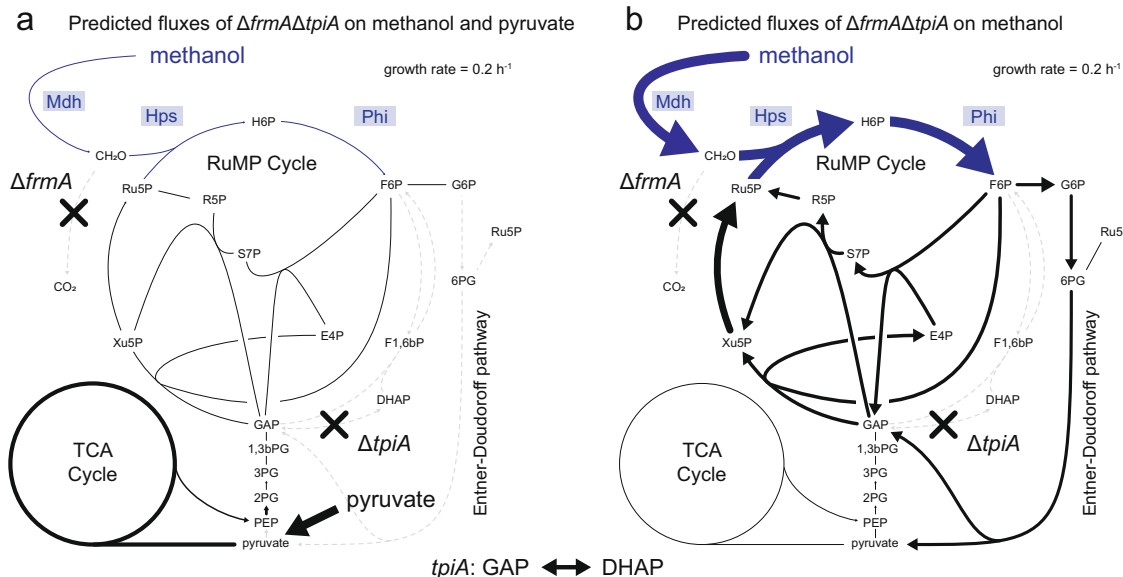

**a** Predicted fluxes of ΔfrmAΔtpiA on methanol and pyruvate  **b** Predicted fluxes of ΔfrmAΔtpiA on methanol

tpiA: GAP ⟷ DHAP

**Fig. 1 | Differences of the flux distribution of the methanol-dependent strain ΔfrmAΔtpiA on methanol and pyruvate and methanol alone.** FVA-predicted average metabolic fluxes in strain ΔfrmAΔtpiA at a set growth rate of 0.2 h$^{-1}$ during growth on methanol and pyruvate as co-substrates (**a**) and methanol as the sole carbon source (**b**) Due to the higher efficiency of *E. coli* to metabolize pyruvate, pyruvate was provided in excess and methanol at limiting concentrations during co-consumption. The minimal amount of methanol required to achieve the growth rate was considered. The TCA cycle flux value originates from the flux through the citrate synthase reaction. Metabolic reactions that were synthetically introduced into the metabolism of *E. coli* and the corresponding enzymes are depicted in blue. Gray, dashed lines indicate zero flux through the reaction. Mdh methanol dehydrogenase, Hps 3-hexulose 6-phosphate synthase, Phi 6-phospho 3-hexuloisomerase, *frmA* S-(hydroxymethyl)glutathione dehydrogenase, *tpiA* triose phosphate isomerase, CH$_2$O formaldehyde, H6P arabino 3-hexulose 6-phosphate, F6P fructose 6-phosphate, G6P glucose 6-phosphate, 6PG 6-phosphogluconate, F1,6bP fructose 1,6-bisphosphate, S7P sedoheptulose 7-phosphate, E4P erythrose 4-phosphate, R5P ribose 5-phosphate, Ru5P ribulose 5-phosphate, Xu5P xylulose 5-phosphate, GAP glyceraldehyde 3-phosphate, DHAP dihydroxyacetone phosphate, 1,3bPG 1,3-bisphosphoglycerate, 3PG 3-phosphoglycerate; 2PG 2-phosphoglycerate; PEP phosphoenolpyruvate; RuMP cycle ribulose monophosphate cycle, TCA cycle tricarboxylic acid cycle. Source data are provided as a Source Data file.

by a factor of four after changing the composition of the feed medium. A gradual increase in optical density was observed over the subsequent 100 generations. Starting from generation 115 until 223, a steady increase in optical density at 600 nm (OD$_{600}$) from 0.23 to 0.70 was observed, indicating that the population was incorporating more methanol. For technical reasons, the bioreactors had to be restarted after 200 and 223 generations of incubation. After another 25 generations and a total of 249 generations, an increase in optical density to 2.5 was observed, indicating another large increase in methanol uptake by the population and potentially the feasibility of growth on methanol as the sole carbon and energy source.

**Growth on and biomass formation from methanol**
The marked increase in yield of the chemostat culture after the long-term evolution of the strain ΔfrmAΔtpiA indicated substantial methanol consumption by the population, which in turn suggested that the population might be capable of growing on methanol also in the absence of pyruvate. When we inoculated the population in a shake flask with medium containing only methanol (500 mM) as a carbon source, the culture was indeed able to grow, reaching an optical density of about 0.4 at a doubling time ($T_d$) of 60 h (Fig. 3a). Interestingly, we noticed that at this point IPTG was not required for methylotrophic growth anymore. To improve the growth performance of the population, we further evolved it under serial transfer conditions in medium containing only methanol as carbon source. After 285 more generations, we isolated and characterized 4 individual clones. The growth rate increased about eightfold in all clones tested ($T_d = 7.5 \pm 0.8$ h (mean ± standard deviation)), while yields were around OD$_{600}$ 2.1 (Supplementary Fig. 1). The best growing isolate (MEcoli_ref_1) was then tested again for growth on methanol ($T_d = 8.1 \pm 0.4$ h (mean ± standard deviation)) and used for all subsequent experiments unless specified otherwise (Fig. 3b).

To confirm complete biomass formation from methanol, we utilized $^{13}$C metabolic tracer analysis and determined the $^{13}$C labeled fraction of protein-bound amino acids by liquid chromatography coupled mass spectrometry (LC-MS) in MEcoli_ref_1 grown in medium containing no additional carbon sources (i.e. no antibiotics, no IPTG, no ethylenediaminetetraacetic acid (EDTA)). When grown under ambient atmosphere, we observed appreciable amounts of residual $^{12}$C label present in protein-bound amino acids (Fig. 4a, b). Due to the absence of additional carbon sources in the medium, the remaining source of $^{12}$C carbon was ambient CO$_2$. Consequently, when MEcoli_ref_1 was grown under an atmosphere enriched to 5% (V/V) $^{13}$CO$_2$, protein-bound amino acids were fully labeled (Fig. 4c, d). A similar labeling pattern was observed in extracted metabolites of methylotrophic clones from an earlier stage of the serial dilution evolution (Supplementary Fig. 2). In addition, we determined the total biomass labeling ratio by elemental analyzer/isotope ratio mass spectrometry. When grown under ambient atmosphere, 83.9 ± 4.6% (mean ± standard deviation) of the total biomass of MEcoli_ref_1 was $^{13}$C labeled. We found that this matches the FBA model that predicts total biomass labeling of about 83% (Supplementary Fig. 3). When MEcoli_ref_1 was grown under $^{13}$CO$_2$ enriched atmosphere 98.6 ± 0.3% (mean ± standard deviation) of its biomass was derived from $^{13}$C, as expected. Lastly, the $^{13}$C fraction of protein-bound amino acids (Fig. 4b) allowed us to discern where CO$_2$ entered metabolism. CO$_2$ did not contribute to the biosynthesis of amino acids directly derived from the RuMP cycle (His, Phe, Tyr) and only little to ones (Ala, Leu, Ser, Val) with gluconeogenic precursors. Noticeably higher levels of CO$_2$ incorporation were found in amino acids derived from the TCA cycle (Arg, Asp, Glu, Ile, Pro, Thr).

**Proteome remodeling in the methylotrophic *E. coli***
Growth on methanol requires large-scale metabolic rewiring, shown above by FVA modeling (Fig. 1). This can occur by fixation of mutations

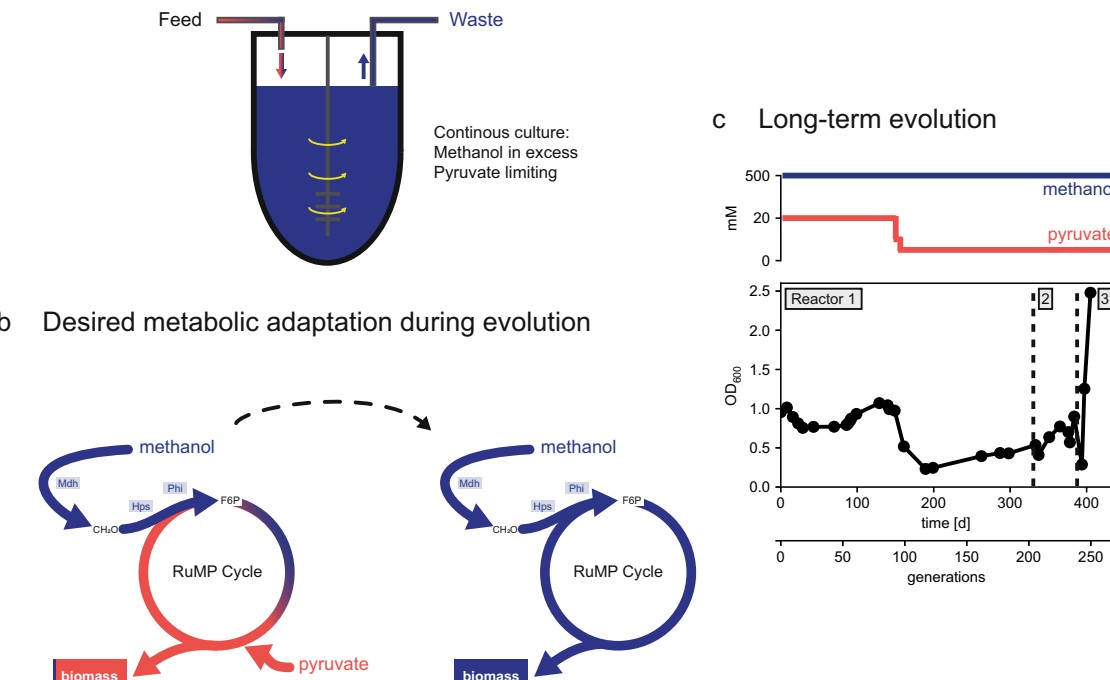

**Fig. 2 | Long-term evolution of the strain ΔfrmAΔtpiA. a** Methanol (blue) and pyruvate (red) in the feed medium were applied in concentrations that ensured an excess of methanol (500 mM) in the bioreactor while pyruvate was kept limiting (< 0.01 mM). **b** Targeted metabolic adaptation of the methanol-dependent strain towards the formation of almost the entire biomass from pyruvate to growth on methanol alone. **c** Experimental boundary conditions, i.e. the concentrations of methanol and pyruvate in the feed medium and the dilution rate of the culture are shown in the top panel and observations in terms of the density of the culture over time and number of generations. Due to technical issues, the reactor 1 was restarted after 200 generations (reactor 2) and reactor 2 after 223 generations (reactor 3). Source data are provided as a Source Data file.

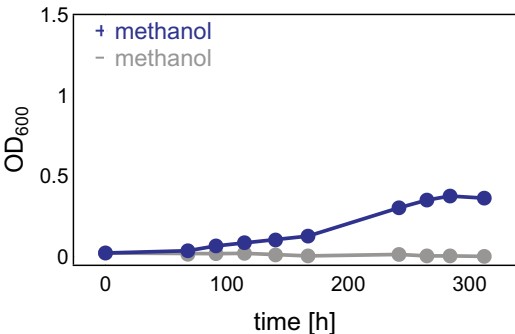

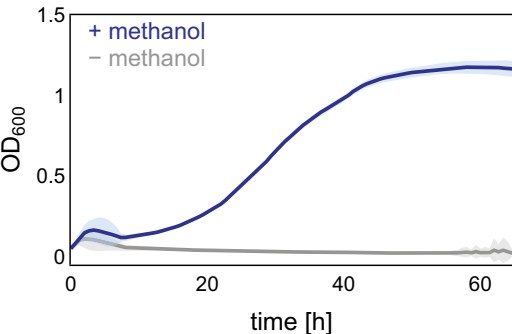

**Fig. 3 | Growth on methanol in the absence of pyruvate of the evolved *E. coli*. a** Growth phenotype of the evolved population after 249 generations in the chemostat, in the presence and absence of methanol. Growth was examined in minimal medium containing 500 mM methanol, ampicillin and streptomycin. **b** Growth phenotype of MEcoli_ref_1 in minimal medium supplemented with 500 mM methanol. Growth was examined by measuring absorbance at 600 nm in a micro-plate reader. The measured absorbance values were converted to $OD_{600}$ values by a calibration curve. Shown are the mean of 10 technical replicates and the standard deviation error band around the mean. Source data are provided as a Source Data file.

that alter individual protein kinetic parameters, by mutations in regulators and promoters or any combination thereof. To observe potential proteome remodeling, we compared the proteome of the ancestral methanol-dependent *E. coli* strain grown on methanol and pyruvate to the proteome of the methylotrophic MEcoli_ref_1 grown on methanol alone. Indeed, about 20% of the detected proteins were differentially expressed between the two strains (1494 detected proteins, differential expression cutoff $|\log_2(\text{fold-change})| \geq 1.5$) (Supplementary Fig. 4, Supplementary Data 1). Overall, enzymes of the RuMP cycle, the Entner–Doudoroff pathway as well as methanol dehydrogenase were upregulated, while enzymes of branch point reactions away from the RuMP cycle, pyruvate metabolism and the TCA cycle were down-regulated (Fig. 5). We further contextualized the proteome data by estimating relative abundances of individual proteins[40], which revealed that methanol dehydrogenase increased in abundance from 16% of the quantifiable proteome in the ancestral strain to 40% in MEcoli_ref_1. This was substantiated by sodium dodecyl sulfate–polyacrylamide gel electrophoresis (SDS-PAGE) (Supplementary Fig. 5).

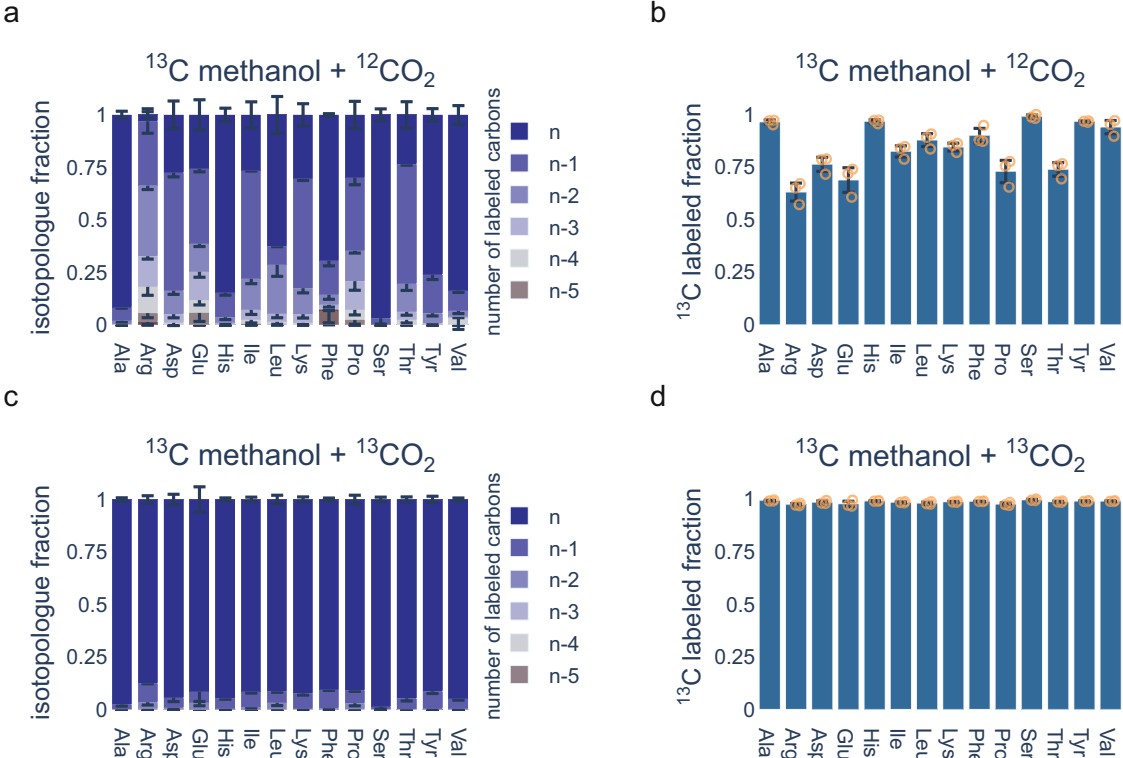

**Fig. 4 | Methanol incorporation into protein-bound amino acids by the evolved methylotrophic _E. coli_ strain.** MEcoli_ref_1 was grown in minimal medium supplemented with 500 mM $^{13}$C methanol either under ambient $CO_2$ (**a**, **b**) or under 5% (V/V) enriched $^{13}CO_2$ atmosphere (**c**, **d**) Antibiotics, IPTG and EDTA were dropped out of the minimal medium to ensure that methanol and $CO_2$ were the only available carbon sources. Isotopologue distributions and fractional contributions were determined for different protein-bound amino acids by LC-MS. For isotopologue distributions, n refers to the fully labeled molecule and _n_-1, _n_-2, etc. to molecules with one, respectively two, etc., unlabeled carbon atoms. Data are represented as mean values ± standard deviation of three replicate inoculations of MEcoli_ref_1. Source data are provided as a Source Data file.

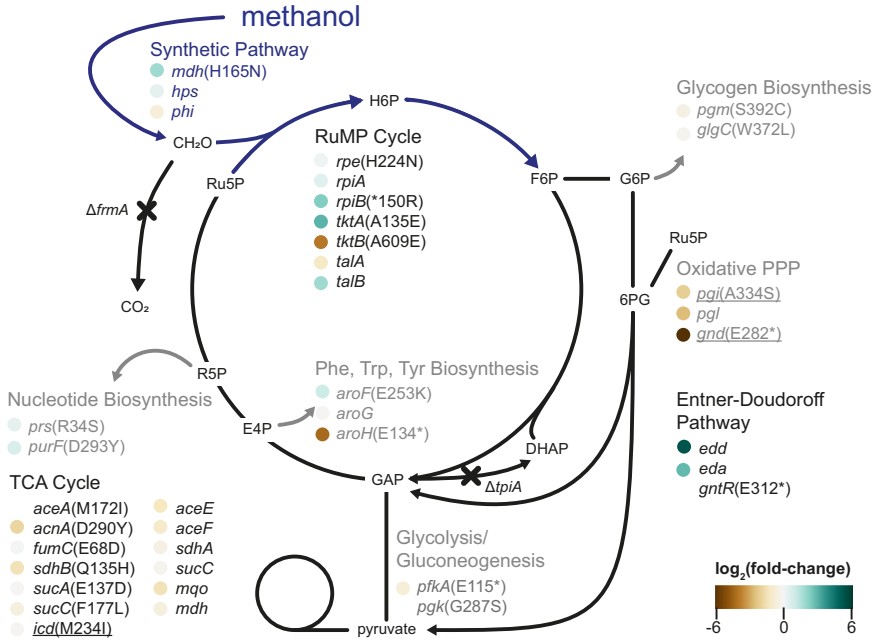

**Fig. 5 | Genetic and proteome changes of methylotrophic _E. coli_ strains.** Mutations in core metabolism present in the methylotrophic population after 249 generations of chemostat evolution are shown. Additional genetic changes found in MEcoli_ref_1 are underlined. Proteomics data, if available for a gene product, are color-coded next to the corresponding gene name. Abbreviations are listed in Supplementary Data 2. Source data are provided as a Source Data file.

To find additional cellular functions with altered expression profiles, we performed gene set enrichment analysis on KEGG pathway annotations. Interestingly, this showed that proteins related to DNA replication, mismatch repair and homologous recombination were overrepresented in the evolved strain (Supplementary Fig. 6).

## Evolutionary trajectory towards methylotrophy and genetic makeup of MEcoli_ref_1

To identify genetic changes acquired during evolution towards methylotrophy, we sequenced the ancestral strain and the methylotrophic reference strain (MEcoli_ref_1). In addition, we determined the metagenomic composition of the evolving population at regular time intervals to obtain temporal information of the adaptation process. After 52 generations the number of observed and fixed (i.e. abundance greater than 90% in the population) mutations started increasing until the end of the experiment (Supplementary Fig. 7a, Supplementary Data 3). This trend was only interrupted at around 100 generations where strains carrying previously highly abundant alleles were temporarily outcompeted by a subpopulation of alternative genetic makeup (i.e. clonal interference) (Supplementary Fig. 7a, c, d). After 249 generations, when the population achieved methylotrophic growth in the chemostat, 564 mutations had fixed in the population. Subsequent evolution under serial dilution regime further increased this number to about 1000. Ultimately, clones isolated after 534 generations, the same ones for which growth rates were determined, acquired on average 1155 ± 52 (mean ± standard deviation) mutations. MEcoli_ref_1 carried 1231 mutations of which 55% (677 of 1231) were nonsynonymous.

The number of genetic changes in MEcoli_ref_1 translates to a time-averaged mutation rate of 2.8 mutations per generation per genome. This is more than 2000-fold higher than the WT *E. coli* mutation rate[41]. To find mutated genes that might have caused the hypermutator phenotype, we searched the sequencing dataset for genes implicated in DNA sequence fidelity that carry fixed nonsynonymous mutations (Supplementary Fig. 7b). Strikingly, a radical amino acid substitution occurred in the conserved exo II motif of *dnaQ*, which encodes the ε-subunit of DNA polymerase III responsible for 3′ → 5′ exonuclease activity[42], early in the evolution.

In an attempt to filter the large amount of nonsynonymous mutations generated by the hypermutator background, we identified mutations in MEcoli_ref_1 that resulted in conservative amino acid replacements and that were previously described in ALEdb[43]. This process yielded only 59 and 1 genomic alterations, respectively, out of 676. Due to the small impact of this filtering step, all further analysis was conducted with the full dataset of observed mutations.

To detect patterns in the evolution towards methylotrophy, we explored the sequence in which metabolic adaptations in methanol dehydrogenase, the RuMP cycle, RuMP cycle efflux points, the TCA cycle and the Entner−Doudoroff pathway arose in the population. Until methylotrophy was achieved, several waves of mutations swept the population. We estimate the number of sweeps between 5 and 9; the definitive number was limited by the temporal resolution of the sampling. Each sweep comprised dozens of mutations and contained some in genes associated with pathways assumed to be important for synthetic methylotrophy in *E. coli* (Supplementary Fig. 7c). Mutations in TCA and RuMP cycle genes dominated the adaptation process during the first 150 generations and were followed by the occurrence of a mutation in methanol dehydrogenase. In the last phase before methylotrophy was achieved, many mutations in the analyzed metabolic pathways fixed in the population. Particularly, the only mutation in the Entner-Doudoroff pathway (*gntR*(E312*)) occurred at this timepoint (Fig. 5 and Supplementary Fig. 7c). Over the course of the serial dilution, another selective sweep occurred, fixing two mutations in RuMP cycle efflux points. Interestingly, despite the higher growth rate of MEcoli_ref_1, the core set of metabolic adaptations was not expanded drastically over continued evolution under serial dilution regime (Fig. 5).

Besides the core methanol metabolization pathways, we searched for other functional classes that were altered during the course of evolution. To this end, we looked for KEGG pathways that were enriched in nonsynonymous mutations in MEcoli_ref_1 but found no significant hits.

Lastly, we tested the impact of the mutation in the key enzyme for methylotrophy, i.e. methanol dehydrogenase. The mutation (H165N) resulted in an about twofold increase in its catalytic turnover number (Supplementary Fig. 8).

## Discussion

Synthetic methylotrophy raised substantial interest in the past years due to the potential of using methanol as a commodity for value-added products in the future and the metabolic engineering challenge to learn which adaptations are required to convert a non-methylotroph into an organism that produces all its biomass from methanol[15−20,23−30,38]. The implementation of the one-carbon assimilation cycle comes with several challenges. Beyond functional expression of the heterologous enzymes, close integration with the native metabolism must be established to allow continuous one-carbon incorporation and, thus, the operation of a synthetic autocatalytic cycle.

In this study, we generated an *E. coli* strain that utilizes the RuMP cycle to grow on methanol as the sole carbon and energy source. We combined rational genetic engineering with long-term laboratory evolution to introduce this complex metabolic trait. In a previous study, we had constructed the strain Δ*frmA*Δ*tpiA* that required methanol for growth[30]. Computational metabolic modeling of the strain indicated that the distribution of metabolic fluxes had to be fundamentally redirected to allow growth on methanol (Fig. 1). Indeed, our laboratory evolution process took about 400 days until methylotrophic growth was observed. Within another 132 days (285 generations) growth on methanol improved substantially to about 8 h. This demonstrates the initial challenge of evolving major transitions to methylotrophy in contrast to improving an already functioning system that requires further optimization. Establishing methylotrophy is inherently harder due to the presence of competing evolutionary pressures. In this case, for example, the utilization of methanol alone or optimization of pyruvate metabolization alongside methanol.

To better understand the evolutionary trajectory of the evolving population, we sequenced it at regular time intervals. Natural selection fixed 1231 unique mutations in the methylotrophic reference strain, MEcoli_ref_1, over the course of 534 generations. On average, it acquired 2.8 mutations per generation per genome which by far exceeds the wild-type mutation rate of *E. coli* of $1 \times 10^{-3}$ mutations per generation per genome[41]. The high mutation rate is likely caused by a mutation in the conserved exo II motif of *dnaQ*, which encodes the ε-subunit of DNA polymerase III responsible for 3′ → 5′ exonuclease activity[42]. Mutations in these domains have previously been shown to result in hypermutators with more than 1000-fold increased mutation rates due to loss of proofreading capability and consequent saturation of the mismatch repair system[42]. The excessively high mutation rate may explain upregulation of DNA repair genes (Supplementary Fig. 4).

A closer investigation of the sequence of metabolic mutations revealed an initial focus on modifications of the RuMP and TCA cycle. This may have primed the *E. coli* metabolism for the subsequent increase in methanol assimilation capability by ensuring sufficient activity of the RuMP cycle for rapid formaldehyde assimilation as well as a balanced intracellular redox state by downregulation of the TCA cycle. The evolution process towards methylotrophy was concluded by activation of the Entner−Doudoroff pathway to link methanol assimilation to lower metabolism, the fixation of mutations in RuMP

cycle efflux points and further changes in RuMP cycle genes to potentially optimize cycle flux.

The breadth of mutations in the evolved *E. coli* suggests that holistic metabolic adaptation is required for growth on methanol as a sole carbon and energy source. Recently, an *E. coli* strain reported by Chen et al. to grow on methanol via the RuMP cycle only acquired 20 mutations during its evolution[18] which differed from the mutations found here. However, the presence of large genome duplications spanning more than 100 kilobases renders proper comparison impossible. In the future, strain reconstruction attempts will help define the minimal set of mutations required for methylotrophic growth of *E. coli*.

The growth of the evolved population and individual clones on methanol as sole source of carbon and energy indicated that all carbon in the biomass originated from methanol. This was confirmed by $^{13}C$ labeling experiments which showed that MEcoli_ref_1 formed $83.9 \pm 4.6\%$ (mean ± standard deviation) of its biomass from $^{13}C$ methanol and the remainder from ambient $CO_2$ through carboxylation reactions, as expected (Fig. 4, Supplementary Fig. 3). To our knowledge, this is the first time that full labeling of biomass was demonstrated in an *E. coli* strain that operates the RuMP cycle.

To enable growth on methanol, MEcoli_ref_1 had to globally rewire its metabolism. In line with this, we observed changes in methanol oxidation, the RuMP cycle, the Entner–Doudoroff pathway, and the TCA cycle.

The ancestral methanol-dependent strain formed about 6.5% of its biomass from methanol. To form all biomass from methanol, our modeling efforts predicted that methanol dehydrogenase activity had to increase substantially (Fig. 1). This could be achieved by increasing enzyme abundance in the cell, in the natural methylotroph *B. methanolicus*[44], for example, methanol dehydrogenase constitutes a large fraction of the proteome, or by expression of an improved enzyme variant. Both strategies were adopted by MEcoli_ref_1. To increase methanol uptake, MEcoli_ref_1 upregulated methanol dehydrogenase about fourfold compared to the ancestral strain (Supplementary Fig. 4) at which level it constituted about 40% of the quantifiable proteome. Additionally, it carried a mutation that doubled the catalytic turnover number (Supplementary Fig. 8). Interestingly, a significant improvement of the enzyme was still possible despite extensive previous engineering efforts[45]. In the future, the evolved methanol dehydrogenase may benefit other synthetic methylotrophy projects due to the central role of the initial methanol oxidation step. For example, it has been suggested that methylotrophic growth via the reductive glycine pathway was limited by methanol dehydrogenase activity[38].

Our modeling efforts predicted enhanced RuMP cycle activity for formaldehyde assimilation into central carbon metabolism. Efficient formaldehyde condensation with ribulose 5-phosphate and, thus, low formaldehyde concentrations, are also required due to the unfavorable thermodynamic equilibrium of the methanol dehydrogenase-catalyzed reaction[26]. MEcoli_ref_1 adapted its proteome accordingly and upregulated three of the seven RuMP cycle enzymes (TktA, TalB, RpiB). Three potentially activity increasing mutations complemented the upregulated enzymes of the RuMP cycle (*rpe, tktA, rpiB*) (Fig. 5). Furthermore, branching points away from the RuMP cycle were downregulated and hit by several mutations. This finding is consistent with theoretical considerations indicating that the Michaelis–Menten parameters of enzymes catalyzing branch-point reactions are critical for the operation of any autocatalytic cycle[31] and is analogous to observations made while evolving an autotrophic *E. coli* strain[32,33,46].

In silico modeling suggested that carbon flux needs to be channeled from the RuMP cycle to the TCA cycle via the Entner–Doudoroff pathway because the glycolytic route was blocked by the *tpiA* deletion in the ancestral methanol-dependent strain. This was mirrored in the evolved methylotrophic *E. coli* strain where the Entner–Doudoroff

pathway repressor, *gntR*[47], carried a pre-mature stop codon (Fig. 5) and the corresponding genes (*edd, eda*) were among the most highly upregulated ones (Fig. 5, Supplementary Fig. 4). The activation of the Entner–Doudoroff pathway is complemented by downregulation and a nonsense mutation in 6-phosphogluconate dehydrogenase (*gnd*) which prevents formation of a futile cycle in which D-gluconate 6-phosphate is converted to ribulose 5-phosphate under loss of $CO_2$.

Accidental loss of carbon equivalents reduces yield and may reduce the growth rate. Indeed, we found that acetyl coenzyme A synthesis from pyruvate (*aceE, aceF*, Fig. 5) was significantly downregulated, suggesting that carbon flux is mainly funneled through phosphoenolpyruvate carboxylase or malic enzyme. This was predicted by our modeling efforts and is metabolically sensible because it enables higher biomass yield per unit methanol than via the acetyl coenzyme A route. Indeed, all TCA cycle-derived amino acids are comprised of about 25% $CO_2$, as expected, since the phosphoenolpyruvate carboxylase/malic enzyme reaction fixes one $CO_2$ for every three carbons from phosphoenolpyruvate and pyruvate, respectively. (Note, arginine shows lower $^{13}C$ labeling because its biosynthesis requires an additional carboxylation step). Interestingly, natural methylotrophs employ the same[48] or similar strategy via pyruvate carboxylase[49].

Lastly, we investigated changes to the TCA cycle. Its main functions are biosynthesis of precursors to major biomass compounds and provision of reduction equivalents. The latter are amply generated by the methanol oxidation reaction, which allows natural methylotrophs to operate downregulated[50,51] or sometimes incomplete TCA cycles[52]. These observations are supported by our in silico analysis and our previous work that shows that reduced TCA cycle activity benefits methanol-dependent growth in *E. coli* presumably by avoiding redox imbalance[29]. MEcoli_ref_1 matched these predictions by lowering expression of TCA cycle genes (Fig. 5). In addition, the strain accumulated several mutations that might have further decreased its activity.

Overall, *E. coli* likely achieved methylotrophic growth by expressing a high-activity methanol dehydrogenase, upregulating the RuMP cycle as well as downregulating efflux points, channeling carbon flux through a carbon-fixing reaction, and avoiding redox-imbalance by running a reduced activity TCA cycle that meets precursor demands rather than generation of redox equivalents. Phenotypically, the methylotrophic *E. coli* adjusted its core metabolism as predicted by modeling studies and theoretical considerations, as shown by our multi-omics characterization. The obtained strain holds great potential for industrial applications and represents the starting point for a platform technology for biotechnological conversion of methanol to numerous value-added compounds.

## Methods

### Reagents and media

Chemicals were obtained from Sigma-Aldrich Chemie GmbH, Buchs, Switzerland unless otherwise specified. The M9 minimal medium used for bacterial cultivation consisted of the following salts (g L$^{-1}$): Na$_2$HPO$_4$ (6.78), KH$_2$PO$_4$ (3.0), NaCl (0.5), NH$_4$Cl (1.0), CaCl$_2$ (0.735), MgSO$_4$ (0.123) and trace elements. Trace elements were present in the medium at the following concentrations (mg L$^{-1}$): Na$_2$EDTA (5.0), MnSO$_4$ (5.0), FeSO$_4$·7H$_2$O (1.0), Co(NO$_3$)$_2$·6H$_2$O (1.0), ZnSO$_4$·7H$_2$O (1.0), CuSO$_4$·5H$_2$O (0.1), Na$_2$MoO$_4$·2H$_2$O (0.1), NiCl$_2$·6H$_2$O (0.2). If indicated, antibiotics were added in the following concentrations (mg L$^{-1}$): ampicillin (100), carbenicillin (50), streptomycin sulfate (20).

### Primers and plasmids

Primers and plasmids used in this study are listed in Supplementary Data 4. The heterologously introduced plasmids were pSEVA424 with the methanol dehydrogenase 2 (*mdh*) variant CT4-1[45] from *Cupriavidus necator* and pSEVA131 with the 3-hexulose 6-phosphate synthase (*hps*)

and the 6-phospho 3-hexuloisomerase (*phi*) from *Methylobacillus flagellatus*[29]. The nucleotide sequence of the plasmids was confirmed by PCR and Sanger sequencing (Microsynth AG; Switzerland). Plasmid maps are available from the Source Data file.

## Strains

The strains used in this study are listed in Supplementary Data 4. The starting strain of the evolution experiment *E. coli* BW25113 Δ*frmA*Δ*tpiA* containing pSEVA424 *mdh C. necator* and pSEVA131 *hps* and *phi M. flagellatus* was described in a previous study[30].

## Flux balance analysis for methanol-dependent and methylotrophic growth

Cobra python (0.20.0) was used to perform flux variability analysis (FVA)[53]. The python version was 3.8.3. The metabolic model used for the analysis was based on the *E. coli* core model with 72 metabolites and 95 metabolic reactions[54]. For a complete representation of the methanol-dependent strain Δ*frmA*Δ*tpiA* that expressed the RuMP cycle genes *mdh*, *hps*, and *phi*, the reactions listed in Supplementary Data 5 were added to the model (-> irreversible; <-> reversible).

Deletions in the triose phosphate isomerase (TPI, dhap_c<-> g3p_c), the *S*-(hydroxymethyl)glutathione dehydrogenase (FRMA, formaldeyhde_c + nad_c -> for_c + nadh_c), and the formate dehydrogenase (FDH, for_c + nad_c -> co2_c + nadh_c) reactions were added before FVA to mirror the genomic background of the strain Δ*frmA*Δ*tpiA*. Furthermore, the NAD-dependent malic enzyme reaction (ME1) was made reversible as it is able to perform its reaction in the carboxylation and decarboxylation direction[55].

The fluxes were predicted for the growth rate of $0.2 \, h^{-1}$ in two conditions: methanol and pyruvate as carbon sources and methanol as the sole carbon source. For both conditions, the minimal methanol amount required to achieve the target growth rate was provided: $0.597 \, mmol \, gCDW^{-1} \, h^{-1}$ for the condition with methanol and pyruvate and $9.119 \, mmol \, gCDW^{-1} \, h^{-1}$ for the condition with only methanol. Pyruvate uptake was set to $10 \, mmol \, gCDW^{-1} \, h^{-1}$. The indicated TCA cycle flux was based on the flux through the citrate synthase reaction. The average flux through the RuMP cycle was calculated from the absolute values of the RuMP cycle reactions methanol dehydrogenase (MEDH), 3-hexulose 6-phosphate synthase (H6PS), 6-phosphate 3-hexuloisomerase (H6PI), ribulose phosphate 3-epimerase (RPE), ribose 5-phosphate isomerase (RPI), transaldolase (TALA), transketolase (TKT1 and TKT2).

## Flux balance analysis of carbon assimilation and dissimilation during methylotrophic growth

Fluxes were predicted for an upper bound of methanol uptake rate of $100 \, mmol \, gCDW^{-1} \, h^{-1}$ while optimizing for growth rate and scaled to $0.0856 \, h^{-1}$ to resemble the growth rate of MEcoli_ref_1 (Fig. 3b). The flux values of carboxylation and decarboxylation reactions were summed up, resulting in the total carboxylation and decarboxylation rates. Considered reactions are listed in Supplementary Data 6.

## Long-term chemostat evolution

The evolution experiment was conducted in a 500 mL bioreactor (Multifors, Infors-HT, Bottmingen, Switzerland) filled with 300 mL minimal medium at 37 °C under constant stirring (700 revolutions per minute (r.p.m.)) and aerated with compressed air. The bioreactor was equipped with medium feed, efflux, acid, and base pump systems. The pH was kept constant at 7.1 by the addition of either hydrochloric acid (HCl) or sodium hydroxide (NaOH). The efflux pump was operated at much higher speed than the feed pump to keep the volume (292, 296, and 299 mL for reactor 1, 2, and 3, respectively) of the culture constant and to maintain the chemostat condition. The culture volume and flow rate were determined by measuring the difference in weight of the feed and waste medium over time. The dilution rates were 0.42, 0.34,

and $0.62 \, d^{-1}$ for reactor 1, 2, and 3, respectively. The feed medium consisted of minimal medium supplemented with 500 mM methanol, 20 mM pyruvate, 0.1 mM IPTG, ampicillin, and streptomycin for the first 90 generations. Then, the pyruvate concentration was reduced to 10 mM and after another 6 generations to 5 mM, respectively. The state of the culture in the chemostat was followed by measuring the optical density of the medium at 600 nm. The chemostat was restarted twice, once after generations 202 generations (contamination) and once after 223 generations (fresh chemostat set at faster dilution rate). The doubling time of the bacterial culture and the number of generations were calculated from standard chemostat equations[56]. At regular intervals, the population was tested for methylotrophic and methanol-dependent growth. To verify that the pyruvate concentration was limiting, sterile-filtered medium aliquots were analyzed by HPLC (UPLC Ultimate 3000, ThermoFisher Scientific, Reinach, Switzerland) equipped with an ion exclusion column (Rezex ROA-Organic Acid H + (8%) 300 × 7.8 mm, Phenomenex, Torrance, CA, United States of America) applying $5 \, mM \, H_2SO_4$ as a mobile phase isocratically. 10 μL of sample were injected at a flow rate of $0.6 \, mL \, min^{-1}$ and the absorbance at 210 nm was recorded for 25 min. The pyruvate concentration was calculated based on a standard curve from samples with known pyruvate levels (0.01, 0.1, 0.5, 1, 5, 10, 20 mM pyruvate) and concentrations below 0.01 mM were considered as below the detection limit.

## Serial transfer evolution

After evolution in a chemostat for 249 generations, the population was propagated under a serial transfer regime. Initially, an aliquot from the chemostat population was passaged seven times in 20 mL minimal medium containing 500 mM methanol, 0.1 mM IPTG, Amp, and Sm. This culture was then used to inoculate four replicate lineages. Each was propagated in 30 mL of the same medium, except that IPTG was omitted, and passaged during mid- or late-exponential phase. All cultures were incubated in 100 mL baffled shake flasks at 37 °C, 160 r.p.m. in a Minitron shaker (Infors-HT, Bottmingen, Switzerland). To inoculate fresh medium, old cultures were diluted 1:100 (V/V).

## Characterization of the growth phenotype after 534 generations

After 534 generations of evolution, the four replicate lineages were streaked out on agar plates containing minimal medium supplemented with 500 mM methanol, ampicillin, and streptomycin. Four colonies were inoculated into 30 mL medium supplemented with 500 mM methanol, ampicillin, and streptomycin and cultivated in baffled shake flasks at 37 °C, 160 r.p.m. in a Minitron shaker. During late-exponential growth, cultures were diluted 1:100 (V/V) in fresh medium of the same composition to assess growth. Growth of the bacterial cultures was monitored by measuring the $OD_{600}$ over time. A cryostock was generated of the fastest growing replicate (MEcoli_ref_1).

## Proteome comparison between ancestral methanol-dependent *E. coli* and MEcoli_ref_1

MEcoli_ref_1 was streaked out on agar plate containing minimal medium supplemented with 500 mM methanol and incubated at 37 °C until colonies were visible. A cross-section of colonies was used to inoculate a pre-culture in 30 mL minimal medium supplemented with 500 mM methanol and cultivated in baffled shake flasks at 37 °C, 160 r.p.m. until stationary phase. Next, the culture was diluted 1:100 (V/V) into fresh medium, grown until mid-exponential phase and split 1:100 (V/V) into five main-culture replicates. Once the cultures reached mid-exponential phase ($OD_{600} \approx 0.6$), 4 OD units (1 OD unit equals 1 mL of culture at $OD_{600}$ of 1) of cells were harvested, cooled to 4 °C, spun down (3220 g, 15 min), and washed once with 4 mL 10 mM $MgCl_2$, and twice with 1 mL 10 mM $MgCl_2$. Finally, the supernatant was discarded, the cell pellet shock frozen in liquid nitrogen and frozen. The same procedure was followed for the ancestral strain

except all media were additionally supplemented with 20 mM pyruvate and 0.1 mM IPTG.

Cell pellets were dissolved in 300 μL 100 mM ammonium bicarbonate, 8 M urea, 1x cOmplete EDTA-free protease inhibitor cocktail (Sigma-Aldrich, Buchs, Switzerland) and lysed by indirect sonication (3 ×1 min, 100% amplitude, 0.8 s cycle time) in a VialTweeter (HIFU, Hielscher, Teltow, Germany). Larger particles and insoluble parts were removed by centrifugation at 13,000 × *g*, 15 min, 4 °C. Protein concentrations in the lysates were determined by Pierce BCA assays (Thermo Fischer Scientific, Reinach, Switzerland). Protein disulfide bonds were reduced by adding tris(2-carboxylethyl)phosphine (TCEP, Sigma-Aldrich, Buchs, Switzerland) to a final concentration of 5 mM and incubation for 30 min at 37 °C and 300 r.p.m. shaking. Cysteine residues were alkylated by adding iodoacetamide (IAA, Sigma-Aldrich, Buchs, Switzerland) to a final concentration of 10 mM for 30 min at room temperature in the dark. Prior to digestion the urea concentration was reduced below 2 M by diluting all samples 1–5 with freshly prepared 50 mM ammonium bicarbonate. For digestion, sequencing grade modified trypsin (Promega AG, Dübendorf, Switzerland) was added at a 1:50 (μg trypsin/μg protein) ratio. Digestion was carried out in a tabletop shaker over night at 37 °C under constant shaking at 300 r.p.m. After digestion, the trypsin was inactivated using heat incubation in a tabletop shaker at 95 °C for 5 min and subsequent acidification to an approximate final concentration of 1% (V/V) formic acid. The peptide samples were centrifuged at 20,000 × *g* for 10 min to remove insoluble parts and the supernatant was taken and desalted using Sep-Pak Vak C18 reversed phase columns (Waters Corporation, Baden-Dättswil, Switzerland) according to manufacturer's instructions and dried under vacuum. Prior to MS analysis, the samples were re-solubilized in 3% acetonitrile (ACN) containing 0.1% formic acid (FA) to a final concentration of 0.5–1 μg μL$^{-1}$.

Mass spectrometry analyses was performed on an Orbitrap Lumos Tribrid mass spectrometer (Thermo Fischer Scientific) equipped with a digital PicoView source (New Objective, Littleton, USA) coupled to an M-Class ultraperformance liquid chromatography (UPLC) system (Waters GmbH, Wilmslow, UK). A two-channel solvent system was used with 0.1% formic acid (V/V) in water for channel A and 0.1% formic acid (V/V), 99.9% ACN (V/V) for channel B. At a peptide concentration of 0.5 μg μL$^{-1}$ for each sample 2 μL were loaded on an ACQUITY UPLC M-Class Symmetry C18 trap column (100 Å, 5 μm; 180 μm x 20 mm, Waters) followed by an ACQUITY UPLC M-Class HSS T3 column (100 Å, 1.8 μm; 75 μm x 250 mm, Waters). Peptide samples were separated at a flow rate of 300 nL min$^{-1}$ with an initial of 5% B for 3 min. The gradient was as follows: from 5 to 22% B in 112 min, from 22 to 32% B in 8 min, from 32 to 95% B in 5 min and from 95 to 5% B in 10 min. The mass spectrometer was operated in data-dependent acquisition (DDA) and full-scan mass spectra were acquired in the Orbitrap analyzer with a mass range of 300–2000 m/z and a resolution of 120k with an automated gain control (AGC) target value of 500,000. Fragment ion spectra (MS/MS) were acquired in the Ion trap using quadrupole isolation with a window of 1.6 Da and fragmented using higher energy collisional dissociation (HCD) with a normalized collision energy of 35%. Only precursor ions with charge states +2 to +7 and a signal intensity of at least 5000 were selected for fragmentation, and the maximum cycle time was set to 3 s. The ion trap was operated in rapid scan mode with an AGC target value of 10,000 and a maximum injection time of 50 ms. The dynamic exclusion was set to 25 s, and the exclusion window was set to 10 p.p.m. Measurements were acquired using internal lock mass calibration on m/z 371.10124 and 445.12003. Sample acquisition was performed in randomized order.

Progenesis QI (Nonlinear Dynamics, v.4.2.7207.22925) was used to process the acquired raw mass spectrometry data. Prior to the automatic alignment, 3–5 vectors were manually seeded to aid the alignment. As alignment reference a 1:1 pool of all samples was used. After normalization, from each peptide ion a maximum of the top five tandem mass spectra were exported. The mascot generic file (*.mgf) was searched using the Mascot server (Matrix Science, v.2.7.0.1) against a decoyed and reversed protein sequence database. Two databases were constructed: For the ancestral strain, one containing the 4449 annotated proteins of the ancestral strain BW25113 (Genbank accession: NZ_CP009273) supplemented with the amino acid sequences of Mdh, Hps, Phi and concatenated with the yeast proteome (Uniprot accession: UP000002311) as well as 260 known mass spectrometry contaminants. For the evolved strain, the same database, but with modified amino acid sequences to account for observed mutations. The Mascot search parameters were as follows: Precursor ion and fragment ion tolerance were set to ±10 ppm and ± 0.5 Da, respectively. Trypsin was selected as protease (two missed cleavages) and ions with charge state 2+, 3+ and 4+ were selected for identification. Carbamidomethylation of cysteine was set as fixed modification and oxidation of methionine, carbamylation of the N-terminus and lysine were set as variable modifications. The mascot search was imported into Scaffold (Proteome Software, v.5.1.0) using 5% peptide and 10% protein false discovery rate (FDR) and the resulting scaffold spectrum report were imported into Progenesis QI. For label-free protein quantification the Hi-3 approach was selected and only proteins with at least two unique peptides were considered for quantification. Statistical testing was performed directly in Progenesis with a one-way ANOVA and the resulting *P* values were adjusted for multiple hypothesis testing using the Benjamini-Hochberg procedure (termed *q* values). General cutoffs for significantly regulated proteins were *q*-values < 0.05 and |log$_2$(fold-changes)| ≥ 1.5.

Relative proteome contributions of individual proteins were estimated using the abundance values obtained from the Hi-3 method in Progenesis QI[40]. Only proteins for which more than three peptides were detected were considered for analysis (i.e. the quantifiable proteome). Next, for each protein the mean peak area was divided by the mean sum of all peaks. This ratio determined the relative protein abundance of a protein.

## Proteomics gene set enrichment analysis of KEGG pathways

Gene set enrichment analysis of the proteomics data was conducted against the KEGG database using clusterProfiler (v.4.4.1, R-version 0.4.2)[57] gseKEGG with the following settings: organism = 'eco', min-GSSize = 3, *p*valueCutoff = 0.05, *p*AdjustMethod = 'BH'.

## Characterization of the growth phenotype of MEcoli_ref_1

One of the replicates of the last proteomics preculture was diluted 1:100 (V/V) into minimal medium supplemented with 500 mM methanol. In the negative control methanol was omitted. The cultures were split into 10 technical replicates of 150 μL each, transferred to a 96 well microtiter plate and incubated at 37 °C, 800 r.p.m. in a Log-Phase 600 reader (Agilent, Basel, Switzerland). Growth was observed by measuring absorbance at 600 nm. A calibration curve was used to convert measured absorbance values to OD$_{600}$ values corresponding to measurements with a pathlength of 10 mm.

## $^{13}$C isotopic tracer analysis of central metabolites

After 368 generations of evolution, the four replicate serial dilution lineages were streaked out on agar plates containing minimal medium supplemented with 500 mM methanol, ampicillin, and streptomycin. Four colonies were inoculated into two different conditions: (1) 30 mL of minimal medium supplemented with 500 mM $^{13}$C methanol (isotopic purity 99%, Euriso-Top GmbH, Saarbrücken, Germany), but without Na$_2$EDTA and without antibiotics in baffled shake flasks, which were incubated at 37 °C, 160 r.p.m. in a Minitron shaker under ambient atmosphere. (2) The same medium, which was sparged with synthetic air (80% (V/V) N$_2$, 20% (V/V) O$_2$) for 30 min before inoculation, in sealed shake flasks with a synthetic air atmosphere containing 5% (V/V) $^{13}$CO$_2$ (99% isotopic purity). In both conditions cultures were incubated at

37 °C, 160 r.p.m. in a Minitron shaker. At late-exponential phase cultures were diluted 1:100 (V/V) into identical conditions.

Once cultures reached optical densities between 0.5 and 1.21, metabolites were extracted from 7 to 10 OD units (1 OD unit equals 1 mL of culture at $OD_{600}$ of 1) of culture by rapid filtration. To this end, 5 OD units of culture were applied onto a 0.2 μm regenerated cellulose filter (RC58, Whatman GmbH, Dassel, Germany), filtered, washed with 10 mL, 37 °C ultrapure water containing 500 mM $^{13}C$ methanol, quenched in 8 mL ice cold acetonitrile/methanol/0.5 M formic acid (60:20:20 (V/V/V)), vortexed for 10 s, and kept on ice for 10 min. Each culture was sampled twice in rapid succession. Following metabolite extraction, samples were lyophilized and subsequently resuspended in 250 μL solvent A/solvent B (90:10 (V/V), see below), centrifuged at 10,000 × g, 4 °C for 10 min, the supernatant transferred to a fresh tube and centrifuges again at 20,000 × g, 4 °C for 10 min and the supernatant transferred into HPLC vials. Assuming 1 OD unit corresponds to 250 μg cell dry weight, each sample was extracted from 2500 μg cell biomass dry weight, except samples from replicate 1 grown under ambient atmosphere and from replicate 4 grown under at 5% (V/V) $^{13}CO_2$, which were extracted from 2250 μg cell biomass dry weight and 1775 μg cell biomass dry weight, respectively, because lower culture volumes were sampled.

Metabolites were analyzed using ultra-high pressure liquid chromatography (UPLC Ultimate 3000, ThermoFisher Scientific, Reinach, Switzerland) equipped with a hydrophilic interaction liquid chromatography (HILIC) column (InfinityLab Poroshell 120 HILIC-Z; 2.1 × 100 mm, 1.9 μm, Agilent Technologies, Basel, Switzerland) coupled to a hybrid quadrupole-orbitrap mass spectrometer (Q Exactive Plus, ThermoFisher Scientific, Reinach, Switzerland). The solvent system consisted of 10 mM ammonium acetate, 7 μM medronic acid in ultra-pure water, pH 9 (solvent A) and 10 mM ammonium acetate, 7 μM medronic acid in acetonitrile/ultra-pure water (90:10 (V/V)), pH = 9 (solvent B)[58]. To separate metabolites, the following gradient was used for elution at a constant flow rate of 500 μL / min: 10% A for 1 min; linearly increased to 40% A over 5 min; 40% A for 3 min; linearly decreased to 10% A over 0.5 min; and held at 10% A for 3.5 min. As setting for the mass spectrometry part of the method, Fourier transform mass spectrometry in negative mode with a spray voltage of −2.8 kV, a capillary temperature of 275 °C, S-lens RF level of 50, an auxiliary gas flow rate of 20, and an auxiliary gas heater temperature of 350 °C was applied. Mass spectra were recorded as centroids at a resolution of 70,000 at mass to charge ratio (m/z) 200 with a mass range of 75–800 m/z and a scan rate of ~4 Hz in full scan mode was used. Of each sample 5 uL were injected.

LC-MS results were analyzed using the emzed framework[59] (emzed.ethz.ch). Metabolite isotopologue peaks were extracted by a targeted approach using commercial standards to define retention time - m/z peak windows applying a m/z tolerance of ± 0.002 Da. In case of the metabolite P5P, a mass tolerance of 0.0015 mass units was used for the analysis. The peak area cut off was set at 20,000 counts s$^{-1}$ μL$^{-1}$ injected. For the metabolite group 2-phosphoglycerate and 3-phosphoglycerate (2PG/3PG), only 3-phosphoglycerate was verified by a commercial standard. Isotopologue fractions ($s_i$) and labeled fraction (LF) were determined as previously described[60] by targeted peak integration of all detected isotopologues utilizing Eq. (1) and Eq. (2) based on m, the abundance of the respective isotopologue; n, the number of carbons in the metabolite of interest; I and j, the isotopologues.

$$s_i = \frac{m_i}{\sum_{j=0}^{n} m_j} \tag{1}$$

$$LF = \frac{\sum_{i=0}^{n} m_i * i}{n * \sum_{i=0}^{n} m_i} \tag{2}$$

Probably due to technical issues, the extracted metabolome of colony 4 grown at enriched $^{13}CO_2$ atmosphere exhibited low to undetectable metabolite concentrations and was not considered for further analysis.

## $^{13}C$ isotopic tracer analysis of protein-bound amino acids and total biomass

Generation of samples for tracer analysis of protein-bound amino acids was identical to ones generated for metabolite analysis. Briefly, MEcoli_ref_1 was streaked out on agar plate containing minimal medium supplemented with 500 mM methanol and incubated at 37 °C until colonies were visible. A cross-section of colonies was used to inoculate a preculture in 30 mL minimal medium supplemented with 500 mM methanol and cultivated in baffled shake flasks at 37 °C, 160 r.p.m. until stationary phase. The preculture was split into three replicates of identical conditions as described above. Cells were harvested between $OD_{600}$ 0.5 and 1.8.

Protein-bound amino acids were isolated following previously established protocols[61]: Cell pellets were resuspended in 200 μL 6 M HCl and baked at 105 °C overnight. The cell hydrolysate was dried at 95 °C under constant airflow, resuspended in 1 mL water and centrifuged twice at 20,000 × g for 10 min to remove insoluble debris. Samples were diluted 1:100 (V/V) in starting conditions of the LC/MS method and analyzed as described above. $^{13}C$ labeling of total biomass samples was conducted by Imprint Analytics (Imprint Analytics GmbH, Neutal, Austria) by elemental analyzer/isotope ratio mass spectrometry.

## Genome resequencing

For genome resequencing, about 2 OD units of cells were sampled, centrifuged for 1 min at 11,000 × g and the supernatant discarded. Genomic DNA was extracted by MasterPure DNA purification kit (Epicenter). Purified genomic DNA was sent for Illumina NovaSeq sequencing (Novogene UK, Cambridge United Kingdom). BBMap (v.38.95) clumpify function was used to filter raw reads for optical and PCR duplicates. The maximum distance to consider for optical replicates was set to the appropriate value for the used sequencer (dupedist = 12,000) and we allowed for one base substitution between duplicates (subs = 1). For PCR duplicates, the same substitution setting was used with two passes for error correction (passes = 2). Filtered reads were aligned to the reference genome of *E. coli* BW25113 (Genbank accession: CP009273) and the plasmid maps of pSEVA424 *mdh2* CT4-1 *Cupriavidus necator* and pSEVA131 *hps phi M. flagellatus* by Breseq (v.0.36.0)[62] in clonal or population mode with default values for all other settings. Genome resequencing samples are summarized in Supplementary Data 3.

## KEGG pathway enrichment of mutations

The set of mutations in MEcoli_ref_1 was analyzed for enrichment of KEGG pathway annotations using clusterProfiler (v.4.4.1, R-version 0.4.2[57] enrichKEGG with the following settings: organism = 'eco', pvalueCutoff = 0.05, qvalueCutoff = 0.2, minGSSize = 3).

## Methanol dehydrogenase activity assay

The ancestral and mutated DNA sequences encoding methanol dehydrogenase were cloned into a pET16b expression vector using Gibson assembly. The resulting constructs added ten histamine residues to the N-terminus of the enzymes for nickel-immobilized metal affinity chromatography purification.

For protein expression both constructs were cultured under the same conditions: A preculture was inoculated in 10 mL LB medium supplemented with carbenicillin and incubated at 37 °C, 160 r.p.m. and diluted 1:50 (V/V) in 400 mL of the same medium in 2 L baffled shake flasks on the next day. The culture was grown to mid-exponential

phase ($OD_{600}$ ~ 0.7) at 37 °C, 160 r.p.m. Once an $OD_{600}$ of about 0.7 was reached, the culture was induced with 0.3 mM IPTG and incubated overnight at 16 °C, 160 r.p.m.

Cells were harvested by centrifugation ($3250 \times g$, 30 min, 4 °C), resuspended in 10 mL lysis buffer (50 mM $NaH_2PO_4$, 300 mM NaCl, 20 mM imidazole, 2 mM dithiothreitol, Roche cOmplete EDTA free protease inhibitor cocktail (Sigma-Aldrich Chemie GmbH, Buchs, Switzerland)) and lysed by sonication (6 mm sonication probe, amplitude 30, process time 4 min, impulse time 5 s, cool down time 15 s, Q700 sonicator (Qsonica LLC, U.S.A.)). Cell debris were cleared by centrifugation ($20,000 \times g$, 40 min, 4 °C). Methanol dehydrogenase was isolated from the resulting solution by fast protein liquid chromatography (ÄKTA, HisTrap HP, GE Healthcare, Chicago, USA) using a linear gradient from starting buffer (lysis buffer without protease inhibitor cocktail) to elution buffer (starting buffer with 500 mM imidazole) over 20 min at a flow rate of 1 mL min$^{-1}$. Buffer was exchanged to reaction buffer (100 mM MOPS, 5 mM $MgSO_4$) by repeated concentration in centrifugal filter units (Amicon Ultra, 10 kDA molecular mass cutoff, Sigma-Aldrich Chemie GmbH, Buchs, Switzerland) and subsequent dilution in reaction buffer until a total dilution factor of greater than 100,000 was achieved.

Methanol dehydrogenase activity was assayed in reaction buffer supplemented with 5 mM nicotinamide adenine dinucleotide ($NAD^+$) as well as 500 mM methanol at 37 °C and by following the formation for NADH/H$^+$, i.e. measuring absorbance at 380 nm in a microplate reader (Tecan Infinite Pro 200, Tecan Group Ltd., Männedorf, Switzerland). Both methanol dehydrogenase variants were added to the reaction mix at equal concentration.

### Reporting summary

Further information on research design is available in the Nature Research Reporting Summary linked to this article.

## Data availability

Data supporting the findings of this work are available within the paper and its Supplementary Data files. A reporting summary for this Article is available as a Supplementary Data file. The *E. coli* core model is accessible from the BIGG FBA model database (http://bigg. ucsd.edu/)[63]. Genome resequencing raw files are available from the sequence read archive (SRA) with BioProject ID PRJNA801133. The accession numbers of samples used for genome sequencing are listed in Supplementary Data 3. The mass spectrometry proteomics data have been deposited to the ProteomeXchange Consortium via the PRIDE partner repository[64] with the dataset identifier PXD034138. Source data are provided with this paper.

## Code availability

Executable scripts and associated data for the generation of all figures are available from the ETH gitlab [https://gitlab.ethz.ch/mreiter/ methylotrophic_ecoli/].

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

## Acknowledgements

We would like to express our gratitude to Jethro Hemmann, Birgitta Ryback and Fabian Meyer for their input and fruitful discussions. We thank Christine Vogel for help with analysis of next generation sequen-cing data and Bernd Roschitzki as well as Jonas Grossmann from the Functional Genomics Center Zurich (FGCZ) for support with LC-MS/MS

setup for proteome analysis. This work was supported by a grant from the Swiss National Science Foundation (310030B-201265) (to J.A.V.).

## Author contributions

Ph.K., M.A.R., Pa.K., E.N., J.A.V. conceived the study. Ph.K. performed FBA/FVA analysis with contributions from E.N. Ph.K., M.A.R., P.C. performed evolution experiments. Ph.K., M.A.R. assessed growth curves. Ph.K., M.A.R., T.G., P.C., Pa.K., performed metabolomics studies. M.A.R. analyzed genome resequencing data and analyzed mutations for enrichment. L.H., M.A.R. performed proteomics studies. Ph.K., M.A.R., J.A.V. wrote the manuscript, with contributions from all authors.

## Competing interests

Ph.K., M.A.R., and J.A.V. filed a patent (application no. EP22185402.9) based on all results presented in this paper. Other authors don't claim competing interests.

## Additional information

**Supplementary information** The online version contains

supplementary material available at

