## [Peer Review File · Nature Communications]

Generation of an Escherichia coli strain growing on methanol via the ribulose monophosphate cycleReviewers' Comments:

Reviewer #1:

Remarks to the Author:

The work of Keller and colleagues is interesting and technically advanced. However it is not clear how the reported work allows additional insights compared to the work of Chen et al. published 2020 (<https://doi.org/10.1016/j.cell.2020.07.010>) Similar approaches were used and similar results were obtained. Additionally, Keller et al. analyzed soluble metabolites to prove isotope incorporation into the biomass. In this case it would be more appropriate to analyze isotope enrichments in e.g. protein bound amino acids (the classical way). The modeling approach is interesting but does not provide further insights, because changes of enzymatic properties caused by genomic alterations were not studied.

Reviewer #2:

Remarks to the Author:

The study of Keller et al., is interesting and seriously made. However, some information and additional experiments are missing to fully appreciate the novelty of this study in the current background where growth of a synthetic methylotroph *E. coli* on pure methanol has already been demonstrated twice in 2020 (Chen et al., 2020 and Kim et al., 2020). Comments that would make the article even more impactful are mentioned below.

Minor:

Line 91: REF should be removed.

Line 163: The concentration of methanol added in the cultivation medium should be mentioned.

Line 167: The unit of the doubling time is missing

Line 186: For which purpose is EDTA added?

L257: The high mutation rates cannot be explained by the presence of the 2 antibiotics used to maintain the plasmids during the evolution? This should be explained in the revised manuscript.

Major:

Line 64: Authors argue that in the study of Chen et al, 2020, a medium with multi-carbon source was used. More details to support this statement should be added in the revised manuscript? More generally I invite the authors to discuss better this paper and the one of Kim et al., 2020 in regard to their results.

Line 164: How to explain that 60 generations of evolution in a shake flask multiplied by five the growth rate while 249 generations in a chemostat were necessary to unlock growth? This should be explained and added in the revised manuscript.

Figure 3B: It is known that maintenance of a plasmid often induces a stress response in *E. coli* resulting in either (i) a folding defect leading to association into protein granules amorphous called inclusion bodies; (ii) proteolysis of the target protein or (iii) reduced growth rate (Mortensen et al., 2005, doi:10.1016/j.jbiotec.2004.08.004). All this leading to a variable activity of the enzymes between the clones expressing the same plasmid. Could it be an explanation for the difference observed between your 3 clones on figure 3B. This should be explained and added in the revised manuscript. This brings me to a more global question: In its state how such a strain can be used in industry?

Figure 4: How can we explain a such big difference on isotopic distribution of arginine pool between the condition labelled/unlabelled CO₂? This should be explained in the revised manuscript.

Line 178, 378 and 385: It seems that no IPTG has been added in the media? If yes why? Does it mean that IPTG is not anymore needed for the expression of *mdh*, *hps* and *phi*? This should be explained and added in the manuscript if true.

Line 385: Why are the evolved strain propagated on minimal M9 methanol medium? Does the clones lost their capacity to growth on pure methanol when passing through LB medium as observed by Chen et al., 2020 ?

In vivo flux experiments on the evolved strain both in stationary condition (using pyruvate plus 13C-

methanol) and in dynamic condition (using pure ¹³C-methanol) would be nice to add in this work. First it will help to confirm both the FBA based results and some metabolic adaptations observed in the evolved strain. Second it would bring more originality to this work in the current background where true synthetic methylotrophy has already been achieved elsewhere by 2 teams. In the same line, a better characterisation of the genomic metabolic adaptation by reintroducing some of the most relevant mutations (e.g. *gntR* and *pgi*) in a wild type background would be nice to validate the benefice of those mutations.

Reviewer #3:

Remarks to the Author:

Building on top of work previously published in Nature Communications, the authors present yet another significant breakthrough towards engineering methylotrophy in *E. coli*. The results are sound and are presented in a well written manuscript.

Flux balance analysis

The use of Flux Balance Analysis (FBA) to show the quantitative metabolic rewiring required to achieve the desired growth is sound and justifies wonderfully the Adaptive Laboratory Evolution (ALE). One caveat is the use of FBA to compare two distributions, it would be more solid to rely on parsimonious FBA or Flux Variability Analysis for this (or maybe Minimization Of Metabolic Adjustment) and we encourage the authors to do so.

A comment on the related methods section Flux balance analysis for methanol-dependent and methylotrophic growth:

- there is an erratum in the reaction 7 ("formaldehyde");
- there were also metabolites added besides reactions (formaldehyde, methanol, hexulose6p and 2ddg6p); and
- the exchange and transport for methanol are missing there.

Also, a table might better fit this information. Why was the core model used for this purpose, instead of, e.g., iML1515?

Adaptive laboratory evolution

The results comply with the initial hypothesis (requirement of a global metabolic rewiring) in that there is a notable enrichment in mutations covering the transcription and translation machinery. This highlights the quantitative nature of this engineering endeavor, where the phenotype might not be explained by some deletions and qualitative information.

Although the achieved phenotype is impressive and a breakthrough, the downstream analysis and the explainability of the results are somewhat underwhelming. First, in terms of the data analysis that was done, I wonder how the enrichment changes when only non-synonymous mutations are considered (is that what is shown?). Second, given the rewiring hypothesis and the results, I was expecting to see transcriptomics or proteomics to support and explain the final phenotype. [Wang et al., 2020](<https://www.ncbi.nlm.nih.gov/pmc/articles/PMC7205612/>) performed an ALE experiment of similar nature that analysed the transcriptomics profile of the strains. Third, given the availability of a lot of ALE sequencing data in <https://aledb.org> I wonder if a comparative analysis could have been conducted to narrow down the very larger number of mutations to a more interpretable set.

Furthermore, the reproducibility of ALE is sometimes fragile because the final results may depend on a particular ordered sequence of mutations. Thus, to enhance the reproducibility and understand the experiment and the final evolved strain, having access to the genetic trajectory followed by the evolving strains would have been important, sequencing the genome at different timepoints of the experiments. This was done, for instance, in [Godara and Kao, 2021](<https://www.ncbi.nlm.nih.gov/pmc/articles/PMC7205612/>). It would have helped to clarify which of the mutations are indeed fixed by genetic drift, whether there were hypermutator

generations (and explain the surprising mutation rate) and the effects of the technical issues in the chemostats, if any. In particular, the enrichment of mutations in antimicrobial resistance genes is problematic, how much of it is improving the fitness, regardless of methylotrophic adaptations? Moreover, not having the genetic trajectory hampers the opportunity to explain the erratic bumps in OD starting from the 200 generation and the genetic cause of the final sharp increase in OD. TL;dr Given that likely one of the next obvious steps will be to try to identify causal mutations from the wealth of mutations in order to introduce them in the background strain, it would have helped to sample and sequence across the evolution experiments and not just at the endpoints.

Other minor comments

- l. 129 To ensure high a -> To ensure a high ...
- l. 142 Maybe the authors could elaborate on the technical issues and also how reactors were restarted in the Methods?

Reviewer #4:

Remarks to the Author:

The same concept was published in 2020 (Chen, et al. Cell, Ref 26 in this manuscript). Although the authors mentioned "However, other multi-carbon sources were present in the medium and it remains open whether the entire biomass was generated from methanol." (Line64-65). Chen et al. achieved growth on methanol as a sole carbon source. Additionally, the strategies and modifications used in this manuscript are very similar with those in Chen et al. Thus, the amount of novel information supplied by the manuscript is not sufficient for publication in this journal.

We thank the four reviewers for taking the time to evaluate our work and for their valuable comments. Below we reply to each suggestion. Line numbers refer to the revised manuscript.

REVIEWER COMMENTS

Reviewer #1 (Remarks to the Author):

The work of Keller and colleagues is interesting and technically advanced. However it is not clear how the reported work allows additional insights compared to the work of Chen et al. published 2020 (<https://doi.org/10.1016/j.cell.2020.07.010>) Similar approaches were used and similar results were obtained. Additionally, Keller et al. analyzed soluble metabolites to prove isotope incorporation into the biomass. In this case it would be more appropriate to analyze isotope enrichments in e.g. protein bound amino acids (the classical way). The modeling approach is interesting but does not provide further insights, because changes of enzymatic properties caused by genomic alterations were not studied.

Thank you for your comments.

In addition to showing labeling data from methanol in free amino acids and central metabolites in our original manuscript, we have now determined the ¹³C labeling also in protein bound amino acids and in the total biomass, as requested. Using both approaches, we confirm that the total biomass is derived from methanol. We have included these data in the updated manuscript (described l. 194-212, Fig. 4).

Regarding the similarity of the approach and results compared to Chen et al., we would like to highlight two fundamental differences between the studies:

i) In the study by Chen and co-workers, the generation of the entire biomass from methanol has not been demonstrated and co-utilization of methanol with additional carbon sources cannot be ruled out from the data shown in the study. This is because the authors determined the labeling in products, i.e. acetate and formate, but not in central metabolites such as citric acid cycle metabolites that serve as precursors for biomass formation or in the biomass itself.

In our experience, biomass labeling of samples can vary from 48 to 85% when cells are grown in the presence of additional carbon sources (ampicillin, streptomycin, IPTG, EDTA). Apparently, *E. coli* can evolve to co-utilize these carbon sources. During evolution, Chen et al. used 50 mM MOPS buffer, a multicarbon compound that can in principle be converted by wildtype *E. coli* (e.g.

<https://doi.org/10.1128/jb.119.3.736-747.1974> and <https://doi.org/10.1074/jbc.272.37.23031>).

Furthermore, morpholine biodegradation does occur in nature

(<https://www.ncbi.nlm.nih.gov/pmc/articles/PMC124687/>). All in all, it is conceivable that *E. coli* evolved to incorporate desulfonated MOPS (3-N-morpholine propanesulfonate) into its biomass. Because the labeling data shown in Chen et al. cannot rule out this scenario, methylotrophic growth (i.e. using methanol as sole carbon and energy source) was not shown.

ii) The evolutionary trajectory of the strain described in our study is clearly different from the study by Chen et al. This is due to a different choice of co-dependency (xylose versus pyruvate and the initial mutant to establish methanol dependency). In our study, we initiated the evolution from an ancestral strain that had the potential to operate a complete ribulose monophosphate pathway (RuMP) from the beginning. In contrast, Chen et al. used a strain with an incomplete RuMP cycle for the ALE experiment, the missing gene was introduced later, followed by continued evolution. This complicates the interpretation of mutations, in addition to genome duplication events observed by the authors.

To gain further insight into the methylotrophic *E. coli* strain, we have now conducted additional characterizations and followed the evolutionary trajectory towards methylotrophy by integrating data from proteomics, genomics and metabolomics (described l. 194-300, Figure 5, Supplementary Figures 2, 4, 5, 6, 7). Among other findings, we examined a point mutation in methanol dehydrogenase that enhances the catalytic turnover of the enzyme (described l. 301, Supplementary Figure 8). This altered methanol dehydrogenase will also benefit other groups in their endeavours to implement alternative methanol-assimilation pathways in non-methylotrophic organisms, all of which rely on the conversion of methanol to formaldehyde as the first catalytic step.

In conclusion, we believe our study provides new insights by demonstrating that a methylotrophic *E. coli* can be established and documenting the occurrence of mutations at the genome scale, and exemplarily validating a promiscuous mutation. We have now also adapted the introduction (l. 70-78) and updated the discussion to better highlight the novelty of our findings and approaches for the reader.

Reviewer #2 (Remarks to the Author):

The study of Keller et al., is interesting and seriously made. However, some information and additional experiments are missing to fully appreciate the novelty of this study in the current background where growth of a synthetic methylotroph *E. coli* on pure methanol has already been demonstrated twice in 2020 (Chen et al., 2020 and Kim et al., 2020). Comments that would make the article even more impactful are mentioned below.

Thank you for your comments and appreciation of our work. We concur with the reviewer that additional experiments improve the impact of our study, and we have now added new data and followed the recommendations of the reviewer as further outlined below.

N.b. We would like to point out that the study by Chen et al. lacks a clear demonstration that a fully synthetic methylotroph was achieved. This is due to the addition of multicarbon compounds in the growth medium and would have required the documentation of labeling data in the total biomass or precursor molecules. We agree that Kim et al., achieved a synthetic methylotroph - in this case using a different assimilation pathway that is less energy efficient and has a lower growth rate than the strain we generated. We adjusted the text to better place our study in the context of previous ones.

Minor:

Line 91: REF should be removed.

We incorporated the change.

Line 163: The concentration of methanol added in the cultivation medium should be mentioned.

The information was added.

Line 167: The unit of the doubling time is missing

We now included the unit of time.

Line 186: For which purpose is EDTA added?

EDTA is added to regular M9 medium as a chelating agent to prevent divalent ions from forming insoluble phosphate salts. However, in our hands, when the medium is freshly prepared, precipitation was not an issue.

L257: The high mutation rates cannot be explained by the presence of the 2 antibiotics used to maintain the plasmids during the evolution? This should be explained in the revised manuscript. The antibiotics were added to the medium to prevent contamination during the long-term evolution experiment. Maintenance of the plasmids is guaranteed by the methanol-dependent growth requirement of the strain. However, we agree that the cause of the high mutation rate deserved better explanation. We have performed an additional analysis that provides an indication to the cause of the hypermutator phenotype. Briefly, we observed a mutation in the ϵ -subunit of DNA polymerase III encoded by *dnaQ* that is involved in its proofreading capability. The mutation was fixed early in the evolutionary process and is the likely cause for the excessively high mutation rate. The manuscript was adjusted accordingly (l. 271-277 and l. 333-342 of the revised manuscript).

Major:

Line 64: Authors argue that in the study of Chen et al, 2020, a medium with multi-carbon source was used. More details to support this statement should be added in the revised manuscript? More generally I invite the authors to discuss better this paper and the one of Kim et al., 2020 in regard to their results. We have now expanded the section concerning the studies by Chen et al. and Kim et al. as requested (l. 70-78).

Line 164: How to explain that 60 generations of evolution in a shake flask multiplied by five the growth rate while 249 generations in a chemostat were necessary to unlock growth? This should be explained and added in the revised manuscript.

Evolving a step change towards methylotrophy is inherently harder because of the presence of competing solutions. In this case, for example, growth on methanol alone vs. optimized growth on pyruvate together with methanol. Once methylotrophy is achieved, optimization of "an unoptimized system" can be more easily selected for: There are plenty of beneficial adaptations available and there is a singular selection pressure. We now discuss the selection pressure and its consequences in the revised manuscript (l. 326-332).

Figure 3B: It is known that maintenance of a plasmid often induces a stress response in *E. coli* resulting in either (i) a folding defect leading to association into protein granules amorphous called inclusion bodies; (ii) proteolysis of the target protein or (iii) reduced growth rate (Mortensen et al., 2005, doi:10.1016/j.jbiotec.2004.08.004). All this leading to a variable activity of the enzymes between the clones expressing the same plasmid. Could it be an explanation for the difference observed between your 3 clones on figure 3B. This should be explained and added in the revised manuscript.

We believe that the differences in growth behavior between individual clones are primarily rooted in genetic differences since these clones were isolated from an evolving population. In the meantime, we continued the evolution and repeated the experiment with more replicates. At this point the population had converged further towards a fitness optimum and hardly any variability was observed (see updated Figure 3B). Furthermore, the reference clone that we now used for omics analyses shows highly reproducible growth over many replicates.

This brings me to a more global question: In its state how such a strain can be used in industry? Biomass formation from methanol represents an ideal starting point to explore applications in the future. However, demonstrating this, is beyond the scope of this study.

Figure 4: How can we explain a such big difference on isotopic distribution of arginine pool between the condition labelled/unlabelled CO₂? This should be explained in the revised manuscript.

It has been shown that arginine biosynthesis requires a carboxylation reaction and thus leads to the incorporation of an additional CO₂ into the molecule (ref. 52-53). Since we use ambient air in the

experiments shown in Fig. 4 a and b, this explains the lower fraction of ^{13}C in contrast to Fig. 4 c and d, where the atmosphere was enriched in $^{13}\text{CO}_2$. We now provide the rationale for the lower labeling of arginine under ambient air containing $^{12}\text{CO}_2$ (l. 403-407).

Line 178, 378 and 385: It seems that no IPTG has been added in the media? If yes why? Does it mean that IPTG is not anymore needed for the expression of *mdh*, *hps* and *phi*? This should be explained and added in the manuscript if true.

This is correct, we added this information to the manuscript (l. 186). Indeed, the promoter is now constitutively active.

Line 385: Why are the evolved strain propagated on minimal M9 methanol medium? Does the clones lost their capacity to growth on pure methanol when passing through LB medium as observed by Chen et al., 2020 ?

This was done routinely as a sanity check to only isolate clones able to grow on methanol as sole carbon and energy source. We tested the robustness of methylotrophy after passaging of the methylotrophic *E. coli* on LB medium for about 20 to 60 generations in LB medium and did not observe a loss of methylotrophy.

In vivo flux experiments on the evolved strain both in stationary condition (using pyruvate plus ^{13}C -methanol) and in dynamic condition (using pure ^{13}C -methanol) would be nice to add in this work. First it will help to confirm both the FBA based results and some metabolic adaptations observed in the evolved strain. Second it would bring more originality to this work in the current background where true synthetic methylotrophy has already been achieved elsewhere by 2 teams.

Thank you for this suggestion. We agree that it would be interesting to perform dynamic labeling experiments. Indeed, we have experience with such experiments but judge these beyond the scope of the study because they require high quantifiability of all mass isotopologues and thus mass resolution, which is intrinsically difficult given the rather small pool sizes. Instead, we were able to discern carbon flux from the RuMP cycle to the TCA cycle using labeling data of protein-bound amino acids using steady state conditions and $^{12}\text{CO}_2$ incorporation from anaplerotic carboxylation reactions (included l. 403). In addition, we now included proteomics data to support the predictions made from the modelling (l. 237-251).

In the same line, a better characterisation of the genomic metabolic adaptation by reintroducing some of the most relevant mutations (e.g. *gntR* and *pgi*) in a wild type background would be nice to validate the benefice of those mutations.

We agree that additional validation experiments would enhance the study. We chose to characterize the mutation observed in *mdh* as the encoded enzyme represents the key step to initiate methylotrophy. We were able to confirm the superior catalytic activity of the enzyme (l. 301, Supplementary Figure 8). Regarding other mutations, we expect that the temporal sequence of introducing these will be important. For example, upregulating the Entner-Doudoroff pathway by inactivation of its repressor *GntR* before the increase of methanol assimilation capacity (improved methanol dehydrogenase) and increased RuMP cycle activity might be counterproductive, as this would deplete RuMP cycle intermediates and may result in growth arrest. This is backed up by our temporal analysis of the evolutionary trajectory of the evolving population in the chemostat which we now added to the manuscript. We anticipate that our thorough analysis of the mutations and their order of appearance represents valuable starting points for testing the effects of mutations beyond *mdh*, which we now included in the study.

Reviewer #3 (Remarks to the Author):

Building on top of work previously published in Nature Communications, the authors present yet another significant breakthrough towards engineering methylotrophy in *E. coli*. The results are sound and are presented in a well written manuscript.

Thank you for the appreciation of the advancement being made in our work.

Flux balance analysis

The use of Flux Balance Analysis (FBA) to show the quantitative metabolic rewiring required to achieve the desired growth is sound and justifies wonderfully the Adaptive Laboratory Evolution (ALE). One caveat is the use of FBA to compare two distributions, it would be more solid to rely on parsimonious FBA or Flux Variability Analysis for this (or maybe Minimization Of Metabolic Adjustment) and we encourage the authors to do so.

Thank you for your comment. Indeed, parsimonious FBA (pFBA) or Flux Variability Analysis (FVA) or potentially Minimization Of Metabolic Adjustment (MOMA) are appropriate to compare the two distributions. We focused on FVA as it provides the solution with the minimal sum of fluxes (as for pFBA), but provides more information about the overall size of the solution space, in particular an estimate of how many different endpoints of flux distribution could lead to growth on methanol during evolution. We decided not to use Minimization of Metabolic Adjustment (MOMA) as the algorithm was designed rather for investigating minor perturbations of central metabolism, but not large-scale rewiring of metabolic networks as required for growth on methanol.

For the FVA analysis, we constrained methanol uptake to satisfy a growth rate of 0.2 h^{-1} , as in the previous analysis. The analysis revealed that for both distributions, i.e. methanol dependent and methylotrophic growth, the solution space was narrow and RuMP cycle reactions exhibited almost no difference between the possible minimum and maximum fluxes. The fluxes (58 out of 114 for methanol/pyruvate, 13 out of 114 for methanol) with a wider range of possible solutions (flux span of more than 5% of the flux center) occur mostly centred around the TCA cycle. Overall, the average fluxes were almost identical to the fluxes obtained by “standard” FBA. Notably, the overall structure of the flux distribution was constant over all FVA solutions. This means that for example the Entner-Doudoroff pathway and RuMP cycle need to be increased in both cases and that flux through the TCA cycle has to be reduced. We integrated the FVA into the manuscript and adapted the corresponding parts accordingly (l. 102-104, 111, 117, Figure 1).

A comment on the related methods section Flux balance analysis for methanol-dependent and methylotrophic growth:

- there is an erratum in the reaction 7 (“formaldehyde”);
- there were also metabolites added besides reactions (formaldehyde, methanol, hexulose6p and 2ddg6p); and
- the exchange and transport for methanol are missing there.

Thank you for your comments related to the FBA method section. We added the missing metabolites and reactions and made the corresponding corrections.

Also, a table might better fit this information. Why was the core model used for this purpose, instead of, e.g., iML1515?

We did not use the genome scale model iML1515 because it includes a large number of side reactions. In our experience, this can lead to solutions based on high fluxes through bypass reactions that are unlikely transferable to in vivo conditions. For example, the deletion of the *tpiA* gene does not lead to methanol-dependent growth on pyruvate in iML1515.

Adaptive laboratory evolution

The results comply with the initial hypothesis (requirement of a global metabolic rewiring) in that there is a notable enrichment in mutations covering the transcription and translation machinery. This highlights the quantitative nature of this engineering endeavor, where the phenotype might not be explained by some deletions and qualitative information.

Although the achieved phenotype is impressive and a breakthrough, the downstream analysis and the explainability of the results are somewhat underwhelming. First, in terms of the data analysis that was done, I wonder how the enrichment changes when only non-synonymous mutations are considered (is that what is shown?). Second, given the rewiring hypothesis and the results, I was expecting to see transcriptomics or proteomics to support and explain the final phenotype. [Wang et al., 2020](<https://www.ncbi.nlm.nih.gov/pmc/articles/PMC7205612/>) performed an ALE experiment of similar nature that analysed the transcriptomics profile of the strains.

We agree that omics analysis provides additional information on the requirements for synthetic methylotrophy. We chose a proteomics approach to have direct access to the relative protein abundances of the ancestral to the evolved strain. Indeed, our data show the strain has undergone major rewiring of its metabolism. We found that both the sequencing data and proteomics data are consistent with predictions from modelling and expectations based on previous knowledge on natural methylotrophy. In addition, we characterized methanol dehydrogenase *in vitro* and found that its catalytic turnover number was increased. We have added the new *in vitro* and *in vivo* proteomics data to the revised manuscript and integrated them with the genomic sequencing data (l. 237-255, Figure 5, Supplementary Figure 4 and 5).

Third, given the availability of a lot of ALE sequencing data in <https://aledb.org> I wonder if a comparative analysis could have been conducted to narrow down the very larger number of mutations to a more interpretable set.

We added this analysis (l. 278-282). The majority of mutations are new and were not previously described in ALEdb.

Furthermore, the reproducibility of ALE is sometimes fragile because the final results may depend on a particular ordered sequence of mutations. Thus, to enhance the reproducibility and understand the experiment and the final evolved strain, having access to the genetic trajectory followed by the evolving strains would have been important, sequencing the genome at different timepoints of the experiments. This was done, for instance, in [Godara and Kao, 2021](<https://www.ncbi.nlm.nih.gov/pmc/articles/PMC7205612/>). It would have helped to clarify which of the mutations are indeed fixed by genetic drift, whether there were hypermutator generations (and explain the surprising mutation rate) and the effects of the technical issues in the chemostats, if any. In particular, the enrichment of mutations in antimicrobial resistance genes is problematic, how much of it is improving the fitness, regardless of methylotrophic adaptations? Moreover, not having the genetic trajectory hampers the opportunity to explain the erratic bumps in OD starting from the 200 generation and the genetic cause of the final sharp increase in OD.

TL;dr Given that likely one of the next obvious steps will be to try to identify causal mutations from the wealth of mutations in order to introduce them in the background strain, it would have helped to sample

and sequence across the evolution experiments and not just at the endpoints.

We agree that time-resolved data on the evolutionary trajectory of the population are interesting and may inform reconstruction efforts in the future. We added this data to the manuscript (l. 257-300, Supplementary Figure 7). Overall, we found that different metabolic pathways appear to be optimized at different timepoints. Regarding the enrichment of antimicrobial resistance genes, we since refined the enrichment analysis and found that all genomic enrichments were spurious. In the hypermutator background most fixed mutations are neutral and hitchhiked to fixation by a few driving mutations.

Other minor comments

- l. 129 To ensure high a -> To ensure a high ...

This has now been fixed

- l. 142 Maybe the authors could elaborate on the technical issues and also how reactors were restarted in the Methods?

We added this information to the revised manuscript.

Reviewer #4 (Remarks to the Author):

The same concept was published in 2020 (Chen, et al. Cell, Ref 26 in this manuscript). Although the authors mentioned "However, other multi-carbon sources were present in the medium and it remains open whether the entire biomass was generated from methanol." (Line64-65). Chen et al. achieved growth on methanol as a sole carbon source. Additionally, the strategies and modifications used in this manuscript are very similar with those in Chen et al. Thus, the amount of novel information supplied by the manuscript is not sufficient for publication in this journal.

We thank for the opportunity to expand on our statement regarding the study by Chen et al. We acknowledge that we need to be more explicit. In the course of our study, we observed that individual samples' total biomass labeling could be as low as 48% when grown in the presence of additional carbon sources (ampicillin, streptomycin, IPTG, EDTA). Apparently, at least some *E. coli* had adapted to co-utilize these. In the Chen et al. study, 50 mM MOPS buffer was used ((Chen et al., p. 935: "Specifically, we grew CFC526.0 in a medium containing methanol and a defined semi-minimal medium, Hi-Def Azure (HDA), that contained amino acids. The HDA amount was sequentially reduced and replaced by a 3-(N-morpholino)propanesulfonic acid (MOPS)-based methanol-minimal medium (MM medium) until the culture could grow on methanol as the sole carbon source", see also Fig. 3, Fig. S5 and Material and Methods section). However, MOPS is a carbon rich compound that can be used as a sulfur source by wildtype *E. coli* (e.g. <https://doi.org/10.1128/jb.119.3.736-747.1974>, <https://doi.org/10.1074/jbc.272.37.23031>). Furthermore, morpholine biodegradation occurs in nature (<https://www.ncbi.nlm.nih.gov/pmc/articles/PMC124687/>). All in all, it is conceivable that *E. coli* evolves to incorporate desulfonated MOPS into its biomass and thus growth on methanol as sole carbon source might not have been achieved. The labeling data shown in Chen et al. cannot rule out this scenario. We added this information to the revised manuscript (l. 70-78). In addition, the evolutionary trajectories towards methylotrophy are clearly different in Chen et al. and our study.

Nonetheless to further substantiate our findings and characterize the strain in more detail, we added several experiments to the revised manuscript: We included proteomics data of the methylotrophic strain, showed protein-bound amino acid and complete biomass labeling from ^{13}C methanol (beyond central metabolites in our original submission), characterized a beneficial mutation in the methanol dehydrogenase as well as a description of the evolutionary trajectory towards methylotrophy. Integrating the data allowed us to observe distinct changes in the metabolism of *E. coli* that enable efficient growth on methanol.

Given the current state of research in the field of synthetic methylotrophy and the substantial additions to the manuscript, we believe our study represents a new milestone in the field.

Reviewers' Comments:

Reviewer #1:

Remarks to the Author:

the authors could answer my questions satisfactory, I have no further concerns.

Reviewer #2:

Remarks to the Author:

All the point raised by the reviewers have been addressed in this revised manuscript. These additions clearly improve the impact of the work. However, the arguments given to claim the novelty of the study in regard to works of Chen et al., 2020 are still not convincing.

First, it is written in the response to reviewer that the MOPS used in the study of Chen et al., 2000 could be used by the evolved strain a source of sulfur as it has been described in the study of Neidhardt et al., 1974. I agree that this could happen but sulfur alone cannot sustain a growth since carbon are needed to build up biomass.

Second, a paper where morpholine (i.e. a derivative of MOPS) is used by Mycobacterium strain is mentioned in the revised manuscript L 73-74 to claim than E. coli could potentially grow on this substrate. But when we are interested by the metabolism, we know perfectly that each bacteria species has its own carbon diet which can overlap but never fully. For instance, B. methanolicus can use glucose as E. coli but the former cannot growth methanol as B. methanolicus does!

Next, if the synthetic methylotroph from Chen et al., 2020 had evolved to co-assimilate additional carbon sources such as 3-(N-morpholino) butanoate as it is mentioned in the revised manuscript L73, I would then expect a growth rate higher than 8.5 hours in view of the number of carbons contains in this molecule.

Finally, the fact that both synthetic methylotrophs obtain here and in the study of Chen et al., 2020 exhibited a similar generation time (i.e. 8 and 8.5 hours respectively) strongly suggest that evolution has only been achieved on methanol in the both cases.

Reviewer #3:

Remarks to the Author:

The authors have adequately addressed all my concerns.

REVIEWERS' COMMENTS

Reviewer #1 (Remarks to the Author):

the authors could answer my questions satisfactory, I have no further concerns.

Thank you.

Reviewer #2 (Remarks to the Author):

All the point raised by the reviewers have been addressed in this revised manuscript. These additions clearly improve the impact of the work. However, the arguments given to claim the novelty of the study in regard to works of Chen et al., 2020 are still not convincing.

First, it is written in the response to reviewer that the MOPS used in the study of Chen et al., 2000 could be used by the evolved strain as a source of sulfur as it has been described in the study of Neidhardt et al., 1974. I agree that this could happen but sulfur alone cannot sustain a growth since carbon are needed to build up biomass.

Second, a paper where morpholine (i.e. a derivative of MOPS) is used by Mycobacterium strain is mentioned in the revised manuscript L 73-74 to claim that *E. coli* could potentially grow on this substrate. But when we are interested by the metabolism, we know perfectly that each bacteria species has its own carbon diet which can overlap but never fully. For instance, *B. methanolicus* can use glucose as *E. coli* but the former cannot grow on methanol as *B. methanolicus* does!

Next, if the synthetic methylotroph from Chen et al., 2020 had evolved to co-assimilate additional carbon sources such as 3-(N-morpholino) butanoate as it is mentioned in the revised manuscript L73, I would then expect a growth rate higher than 8.5 hours in view of the number of carbons contained in this molecule.

Finally, the fact that both synthetic methylotrophs obtained here and in the study of Chen et al., 2020 exhibited a similar generation time (i.e. 8 and 8.5 hours respectively) strongly suggest that evolution has only been achieved on methanol in the both cases.

Thank you for your comments. We think there is a misunderstanding. We agree that sulfur alone cannot sustain growth. We also agree that *E. coli* would need to evolve to grow on MOPS. We feel it is difficult to predict potential growth rates in such a case. To avoid ambiguity in detailing differences on media composition during the long-term evolution in the study by Chen et al. compared to ours, we removed the statement that we had added upon revision of the manuscript.

Reviewer #3 (Remarks to the Author):

The authors have adequately addressed all my concerns.

Thank you.